# Revisiting Graph Adversarial Attack and Defense From a Data Distribution Perspective

**Kuan Li, Yang Liu, Xiang Ao,** *Qing He
Key Lab of Intelligent Information Processing of Chinese Academy of Sciences
Institute of Computing Technology, Chinese Academy of Sciences
University of Chinese Academy of Sciences
likuan_buaa@163.com, liuyang520ict@gmail.com, {aoxiang, heqing}@ict.ac.cn

## Abstract

Recent studies have shown that structural perturbations are significantly effective in degrading the accuracy of Graph Neural Networks (GNNs) in the semi-supervised node classification (SSNC) task. However, the reasons for the destructive nature of gradient-based methods have not been explored in-depth. In this work, we discover an interesting phenomenon: the adversarial edges are not uniformly distributed on the graph, and a majority of perturbations are generated around the training nodes in poisoning attacks. Combined with this phenomenon, we provide an explanation for the effectiveness of the gradient-based attack method from a data distribution perspective and revisit both poisoning attack and evasion attack in SSNC. From this new perspective, we empirically and theoretically discuss some other attack tendencies. Based on the analysis, we provide nine practical tips on both attack and defense and meanwhile leverage them to improve existing attack and defense methods. Moreover, we design a fast attack method and a self-training defense method, which outperform the state-of-the-art methods and can effectively scale to large graphs like ogbn-arxiv. We validate our claims through extensive experiments on four benchmark datasets.

## 1 Introduction

Graph Neural Networks (GNNs) have been widely explored in recent years for numerous graph-based tasks Li et al. (2015); Kipf & Welling (2017); Hamilton et al. (2017); Liu et al. (2021b), primarily focused on the semi-supervised node classification (SSNC) task Xu et al. (2019b); Veličković et al. (2017); Huang et al. (2022); Liu et al. (2022). The evidence that GNNs are vulnerable to adversarial structure perturbations is convincing Dai et al. (2018); Zügner et al. (2018); Zügner & Günnemann (2019); Wu et al. (2019); Geisler et al. (2021); Zhu et al. (2022b). Attackers can degrade classification accuracy largely by unnoticeably modifying graph structure. Most attack methods are gradient-based Chen et al. (2020); Wu et al. (2019); Zügner & Günnemann (2019); Xu et al. (2019a); Geisler et al. (2021), treating the adjacency matrix as a parameter and modifying it via the gradient of the attack loss. However, we still lack a general framework to explain their effectiveness.

We posit that the destructive power of gradient-based methods stems from their ability to effectively increase the distribution shift between training nodes and testing nodes.To illustrate in more detail, we start with an interesting phenomenon: the malicious modifications generated by gradient-based methods are not uniformly distributed on the graph. As shown in Fig. 1, most modifications are around the training nodes (ordered at the top of the adjacency matrix), while the largest part of the graph, Test-Test, is hardly affected. Specifically, we apply two representative attack method, MetaAttack Zügner & Günnemann (2019) and PGD Xu et al. (2019a). The data split follows 10%/10%/80% (train/validation/test). Furthermore, we find that only MetaAttack can adaptively adjust the attack tendency (attack training nodes or testing nodes) according to the size of the training set, and such adaptivity makes MetaAttack outperform other methods regardless of the data split. It inspires us to study the effectiveness of attack methods from another perspective, **Distribution Shift**, which likewise considers the differences between the training set and the testing set. This begs the

---

*Corresponding to Xiang Ao

| Attack | Ptb rate | Train-Train | Train-Test | Test-Test |
|---|---|---|---|---|
| | 5% | 13 | 238 | 2 |
| MetaAttack | 10% | 60 | 442 | 3 |
| | 15% | 69 | 673 | 17 |
| | 20% | 79 | 919 | 15 |
| | 5% | 70 | 172 | 3 |
| PGD | 10% | 122 | 365 | 8 |
| | 15% | 162 | 575 | 10 |
| | 20% | 241 | 755 | 12 |

Figure 1: **Left**: The adjacency matrix of Cora attacked by MetaAttack, in which the blue pots are original edges, and the reds are adversarial edges. The green dotted line is the boundary of training nodes and testing nodes. **Right**:The location statistics of adversarial edges on the Cora dataset under different perturbation rates. Train-Train means the perturbed edge links two nodes from the training set. Train-Test and Test-Test follow the same rule.

following challenge:

*How to formulate the distribution shift in graph adversarial attack scenario?*

To answer the above question, in this paper, we first clarify the differences between attack in the mainstream domain, e.g., image classification, and in SSNC: (1) SSNC is a transductive task in which attackers have access to both training nodes and testing nodes; (2) Nodes in the graphs have both features and structural information, while other data types may only contain features, e.g. images have pixels or text have words. Taking these two differences into account, we provide a formalization of the distribution shift in graph adversarial attack and theoretically prove that the perturbations around training nodes enlarge the distribution shift in an effective way. we explore factors that influence the location of adversarial edges, such as surrogate loss and gradient acquisition.

Using the formulation of the distribution shift, some unexplained phenomena in previous works become clear. For example, why do gradient-based attack methods significantly outperform the heuristic homophily-based method, DICE? Why do most modifications are insertions instead of deletions? We will analyze them from both theoretical and empirical sides.

With the advantage of the above analysis, several practical tips are proposed to improve and instruct attack and defense on graphs. To validate these tips, we conduct extensive corresponding experiments. Additionally, we design a fast and straightforward heuristic attack method underpinned by increasing the distribution shift, and it achieves comparable performance to gradient-based methods and can effectively scale to large graphs like ogbn-arxiv. We also provide a self-training-based method to improve the robustness of GNNs[1]. The codes are available at https://github.com/likuanppd/STRG.

Our main contributions are summarized below:

- We find an interesting phenomenon that perturbations are unevenly distributed on the graph, and it inspires us to revisit graph adversarial attack from a data distribution perspective and define the distribution shift in graph attack scenario.

- We explore some unexplained phenomena and provide relevant theoretical proofs from the view of data distribution. We argue that the effectiveness of the graph attack essentially comes from increasing the distribution shift, as the fundamental nature of the adversarial attack.

- We provide some practical tips to instruct both attack and defense on graphs. We conduct extensive experiments to support our claims and verify the validity of these tricks. The implementation details and the statistics of the datasets are provided in A.1.

## 2 RELATED WORK

Many efforts have been made to study various properties of the gradient-based attack algorithms. GCN-SVD Entezari et al. (2020) discovers that attacks exhibit a specific behavior in the spectrum

---

[1]Our focus is to revisit both attack and defense sides from a new view. These two algorithms are natural byproducts of this work, so we put them in the appendix.

of the graph: only high-rank (low-valued) singular components of the graph are affected. Likewise, Chang et al. (2021) and Chang et al. (2022) also study the robustness of GNNs from the spectral perspective. Chang et al. (2021) indicates that not all low-frequency filters lick GCNs are robust to adversarial attacks and proposes GCN-LFR which enhance the robustness of various kinds of GCN-based models by a general robust co-training paradigm. Meanwhile, many efforts have been devoted to revealing the various properties of gradient-based attacks. Zügner & Günnemann (2019) demonstrates that the perturbations tend to increase the heterophily of the graph. Based on it, Zhang & Zitnik (2020) and Wu et al. (2019) invent GNNGuard and Jaccard, respectively, to prune the edges that link two dissimilar nodes. Geisler et al. (2020) and Chen et al. (2021) study the robustness of GNNs from the breakdown point perspective and propose more robust aggregation approaches. Xu et al. (2022) introduce a mutual information-based measure to quantify the robustness of graph encoders on the representation space. Zhan & Pei (2022) is another work that finds the uneven distribution of perturbations, mainly focusing on the attacking side and proposing a black-box attack method. Different from them, we prefer to study the mechanism behind it, including the reasons for its generation and the impact on the effectiveness of the attack methods. We fomulate the distribution shift in graph adversarial attack and leverage it to analyze other tendencies of gradient-based attack methods, which can provide some theoretical guidance and help us understand the robustness and vulnerability of GNNs. Meanwhile, several tips are proposed, covering all the structure attack aspects, and most of them are not mentioned in Zhan & Pei (2022).

## 3 PRELIMINARIES

**Notations.** Let $\mathcal{G} = \{\mathcal{V}, \mathcal{E}\}$ denote an undirected, unweighted graph with $N$ nodes, where $\mathcal{V}$ and $\mathcal{E}$ (without self-loops) are the sets of nodes and edges, respectively. The topology of the graph can also be represented as a symmetric adjacency matrix $\mathbf{A} \in \{0, 1\}^{N \times N}$, in which $\mathbf{A}_{ij} = 1$ denotes that node $v_i$ connects node $v_j$, otherwise $\mathbf{A}_{ij} = 0$. The original features of all nodes can be summarized as a matrix $\mathbf{X} \in \mathbb{R}^{N \times d}$. The first-order neighborhood of node $v_i$ is denoted as $\mathcal{N}_i$, including node $v_i$ itself. Moreover, the labels of all nodes are denoted as $\boldsymbol{y}$. Each node is associated with a label $y_i \in \mathcal{C}$, where $\mathcal{C} = \{c_1, c_2, ..., c_K\}$. We use $f_\theta(\mathbf{A}, \mathbf{X})$ to denote a GNN, and $\theta$ refers to the parameters.

**SSNC.** In this paper, we study the robustness of GNNs on the semi-supervised node classification (SSNC) task, which can be formulated as this: Given a graph $\mathcal{G}$, the node features $\mathbf{X}$, and a subset of node labels $\boldsymbol{y}_L \subset \boldsymbol{y}$, the goal is to learn a function: $\mathcal{V} \to \mathcal{C}$ which maps the nodes to the label set so that we can predict labels of unlabeled nodes. The ground truth labels of unlabeled nodes are denoted as $\boldsymbol{y}_U$, and the corresponding node set is $\mathcal{V}_U$.

**Graph Adversarial Attacks** In this paper, we explore the robustness of GNNs on SSNC in gray-box non-targeted attack on graph structure. Under this setting, the attacker possesses the same data information as the defender, but the defense model and its trained weights are unknown. The adversarial attack can be divided into two categories, namely *poisoning* (training time) attack and *evasion* (testing time) attack Zügner et al. (2018). The attacker aims to find an optimal perturbated graph $\hat{\mathcal{G}}$ that degrades the overall performance of the classifier as much as possible, which can be formulated as Zügner & Günnemann (2019); Geisler et al. (2021):

$$\underset{\hat{\mathbf{A}} \in \Phi(\mathbf{A})}{\arg\min} \quad \mathcal{L}_{atk}(f_{\theta^*}(\hat{\mathbf{A}}, \mathbf{X}), \boldsymbol{y}), \tag{1}$$

where $\hat{\mathbf{A}}$ is the adjacency matrix of the purturbed graph $\hat{\mathcal{G}}$, and $\Phi(\mathbf{A})$ is a set of adjacency matrix that fits the unnoticeable constraint: $\frac{\|\hat{\mathbf{A}} - \mathbf{A}\|_0}{\|\mathbf{A}\|_0} \leq \Delta$, in which $\Delta$ is the maximum perturbation rate. $\mathcal{L}_{atk}$ is often $-\mathcal{L}(f_{\theta^*}(\hat{\mathbf{A}}, \mathbf{X})_U, \hat{\boldsymbol{y}}_U)$ or $-\mathcal{L}(f_{\theta^*}(\hat{\mathbf{A}}, \mathbf{X})_L, \boldsymbol{y}_L)$, where $\hat{\boldsymbol{y}}_U$ is the pseudo-label of unlabeled nodes predicted by the surrogate classifier. For notation simplicity, we call them $\mathcal{L}_{self}$ and $\mathcal{L}_{train}$, respectively. The $\theta^*$ refers to the parameters of the surrogate GNN, which is different in evasion and poisoning attack. It is fixed and trained on the clean graph in evasion attack, but it can be repeatedly retrained as the graph is gradually contaminated in poisoning attack.

The key to the gradient-based attack methods is to treat the adjacency matrix as a parameter and modify the graph structure via the gradient of the attack loss $\nabla_{\mathbf{A}} \mathcal{L}_{atk}$.

## 4 RETHINKING GRAPH ADVERSARIAL ATTACKS

### 4.1 THE LOCATION OF ADVERSARIAL EDGES

To better understand which part of the graph is perturbed, we propose an intention score (**IS**) to quantify how the adversarial edges are distributed on the graph. Since GNN models are permutation invariant Zou et al. (2021), we can adjust the node index and then split the adjacency matrix into the following form:

$$\mathbf{A} = \begin{bmatrix} \mathbf{A}_1 & \mathbf{A}_2 \\ \mathbf{A}_2^\top & \mathbf{A}_3 \end{bmatrix}, \ \mathbf{A}_1 \in \mathbb{R}^{N_1 \times N_1}, \ \mathbf{A}_2 \in \mathbb{R}^{N_1 \times N_2}, \ \mathbf{A}_3 \in \mathbb{R}^{N_2 \times N_2}, \tag{2}$$

where $\mathbf{A}_1$ is an matrix of the edges that connects two training nodes, $\mathbf{A}_2$ is a matrix that represents edges between a training node and a testing node, and $\mathbf{A}_3$ is a matrix where the element denotes an edge between two testing nodes. $N_1$ and $N_2$ are the numbers of the nodes in the training set and testing set.

**IS** can describe in which part of the adjacency matrix ($\mathbf{A}_1$, $\mathbf{A}_2$, or $\mathbf{A}_3$) the attack algorithm prefers to generate adversarial edges. It is be formulated as:

$$\pi_i = \frac{\left| \tilde{\mathcal{E}}_i \right|}{\left| \tilde{\mathcal{E}} \right|}, \ \lambda_i = \frac{|\mathcal{E}_i|}{|\mathcal{E}|}, \ \mathbf{IS}_i = \frac{\pi_i}{\lambda_i}, \ i = 1, 2, 3 \tag{3}$$

where $\mathcal{E}$ is the original edge set, $\tilde{\mathcal{E}}$ denotes the edges inserted or deleted by the attack model, and $i$ denotes the part of the adjacency matrix. **IS** can demonstrate the density of the perturbations located in the adjacency matrix (see Table 1, Table 4, and Fig. 5)

Table 1 shows the results on dataset Cora attacked by two poisoning method, MetaAttack and PGD, under different data split. *MetaAttack will adaptively adjust the attack tendency according to the size of the training set*. When the training size is small, the algorithm tends to modify the local structure around training nodes. The $\mathbf{IS}_3$ increases as the training set becomes larger, meaning that adversarial edges are more likely to be generated around testing nodes. When the size of the training set is 50%, all the **IS** are close to 1, and meanwhile the adversarial edges are nearly uniformly distributed on the graph. More results of other attack methods and on more datasets can be seen in Table 4 in the appendix A.2.

We find that only MetaAttack has such adaptivity, and it make MetaAttack extremely effective in poisoning attack. Other representative gradient-based attack methods, like PGD and FGSM, will always focus on modifying the local structure of the training nodes and only attack the training nodes regardless of the size of the training set. This adaptivity essentially differentiates between training and testing nodes when generating attacks. It motivates us to study this problem from another perspective that likewise considers the differences between the training and testing sets–**Distribution Shift**.

Table 1: The **IS** on Cora dataset under 10% perturbation rate.

| Training set | MetaAttack | | | | PGD | | | | Clean |
| | Train-Train (IS$_1$) | Train-Test (IS$_2$) | Test-Test (IS$_3$) | Acc | Train-Train (IS$_1$) | Train-Test (IS$_2$) | Test-Test (IS$_3$) | Acc | |
| --- | --- | --- | --- | --- | --- | --- | --- | --- | --- |
| 10% | 11.881 | 4.862 | 0.007 | 71.62 | 22.178 | 4.279 | 0.002 | 77.92 | 83.56 |
| 20% | 4.200 | 2.513 | 0.042 | 80.35 | 6.832 | 2.234 | 0.009 | 82.32 | 85.67 |
| 30% | 2.042 | 1.605 | 0.290 | 81.27 | 4.444 | 1.414 | 0.008 | 84.56 | 86.99 |
| 40% | 1.762 | 1.136 | 0.546 | 81.54 | 3.057 | 1.056 | 0.000 | 86.23 | 87.01 |
| 50% | 0.798 | 1.150 | 0.901 | 83.05 | 2.107 | 0.931 | 0.008 | 86.91 | 87.22 |
| 60% | 0.642 | 1.182 | 1.260 | 83.11 | 1.711 | 0.738 | 0.000 | 88.03 | 88.04 |
| 70% | 0.625 | 1.299 | 1.625 | 81.92 | 1.519 | 0.585 | 0.000 | 89.55 | 89.98 |
| 80% | 0.383 | 1.933 | 3.409 | 80.46 | 1.160 | 0.755 | 0.001 | 88.88 | 88.87 |
| 90% | 0.364 | 3.426 | 8.893 | 79.38 | 1.107 | 0.473 | 0.000 | 93.54 | 93.69 |

### 4.2 DISTRIBUTION SHIFT IN GRAPH ATTACKS

Machine learning models cannot perform well when the training distribution is far from the testing distribution Wiles et al. (2021). Intuitively, the smaller the training set, the easier the training

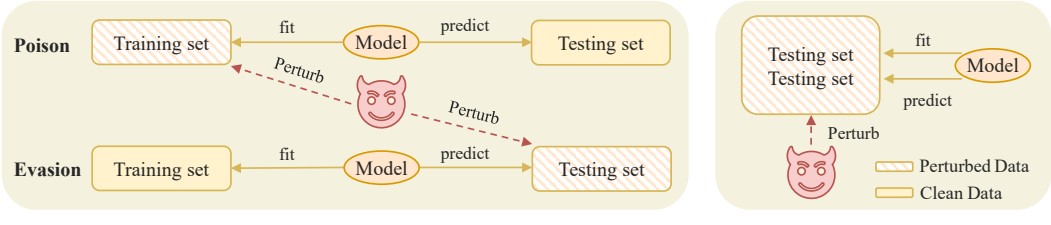

(a) Image Classification

(b) Node Classification

Figure 2: (a): The training set and the testing set are separated in inductive learning under full supervision. For poisoning attack, attackers perturb the training set to fool the classifier. Then the biased model cannot precisely predict the testing set. For evasion attack, the model trained on clean data will be tested on the perturbed data. (b) In SSNC, training nodes and testing nodes co-exist in the graph. Whether it is poisoning attack or evasion attack, the attackers have access to all nodes.

distribution is to be perturbed. The destructive power of these gradient-based methods may stem from the fact that they effectively increase the distribution shift. The first thing we need to do is to formulate the distribution shift in graph adversarial attack.

Unlike attacks in other domain, like image classification, formulating the distribution shift in SSNC should consider the structural information. We assume all the node features are sampled from $p(\boldsymbol{x}|y)$ amd define a community feature to take structural information into account: $\tilde{\boldsymbol{x}}_i = \frac{1}{|\mathcal{N}_i|} \sum_{j \in \mathcal{N}_i} \boldsymbol{x}_j$ and assume the community feature of arbitrary node $i$ follows a distribution:

$$\frac{1}{|\mathcal{N}_i|} \sum_{j \in \mathcal{N}_i} \boldsymbol{x}_j \sim p(\tilde{\boldsymbol{x}}|y). \tag{4}$$

The first-order neighborhood plays the most important role in structure information, so we do not consider higher-order here. All the community features and corresponding labels can be viewed as being sampled from the joint distribution $p(\tilde{\boldsymbol{x}}, y)$ on clean graphs. In graph adversarial attack, there are three key distributions–$p_{train}(\tilde{\boldsymbol{x}}, y)$, $p_{test}(\tilde{\boldsymbol{x}}, y)$, and the classifier $p_\theta(y|\tilde{\boldsymbol{x}})$. In the clean graphs, we assume that $p_{train}(\tilde{\boldsymbol{x}}, y)$ and $p_{test}(\tilde{\boldsymbol{x}}, y)$ are the same as the true distribution $p(\tilde{\boldsymbol{x}}, y)$. In the perturbed graphs, we consider attacking the structures as perturbing the corresponding distribution. Concretely, we treat the structural changes to the training and test nodes as if these nodes were sampled from biased $p_{train}$ and $p_{test}$.

In image classification, the attackers degrade the performance of the classifier by perturbing $p_{train}$ or $p_{test}$. When implementing the poisoning attack, the $p_{train}$ will be perturbed, and the classifier will fit a biased distribution so that it will fail to precisely predict the unbiased images sampled from $p_{test}$. In evasion attack, the attackers have no access to the training set, so they perturb the testing images–this can be viewed as predicting on a biased distribution $p_{test}$ with an unbiased classifier.

However, SSNC is semi-supervised. Attackers can modify the entire graph structure, including the training and test nodes. The difference between SSNC and image classification is shown in Fig. 2. Unlike inductive learning under full supervision scenario, $p_{train}$ and $p_{test}$ can be simultaneously perturbed.

Awareness of this difference is critical. For example, in the image poisoning attack, attackers can only perturb the training data to bias the model. In contrast, in the graph poisoning attack, attackers can perturb the testing data to make an unbiased model test on the biased data.

After perturbing, $p_{train}$, and $p_{test}$ might be different to the true distribution $p(\tilde{\boldsymbol{x}}, y)$ and this discrepancy may result in a distribution shift. By a simple factorization, we can write:

$$p(\tilde{\boldsymbol{x}}, y) = p(y)p(\tilde{\boldsymbol{x}}|y) \tag{5}$$

Labels are not flipped in the setting of structure attack, so we assume $p(y)$ is shared across all distributions. The distribution shift arises when $p_{train}(\tilde{\boldsymbol{x}}|y)$ and $p_{test}(\tilde{\boldsymbol{x}}|y)$ differ due to the structural perturbations, so we define the distribution shift in graph adversarial attack as:

$$\frac{1}{|\mathcal{C}|} \sum_{y \in \mathcal{C}} D_{KL}(p_{train}(\tilde{\boldsymbol{x}}|y = c_i), \ p_{test}(\tilde{\boldsymbol{x}}|y = c_i)) \tag{6}$$

Essentially the attack is to increase the distribution shift, but specifically to perturb $p_{train}(\tilde{\boldsymbol{x}}|y)$ or $p_{test}(\tilde{\boldsymbol{x}}|y)$ in the poisoning attack and evasion attack is different.

For evasion attack, perturbing $p_{train}$ is nearly invalid. The classifier is already trained, which can be viewed as an unbiased model. It is unwise to waste limited modifications on the training nodes. For poisoning attack, the implications of attacking the training and testing sets are different from a distribution perspective. Attacking the training set is to perturb the $p_{train}(\tilde{\boldsymbol{x}}|y)$ in such a way that the classifier $p_\theta(y|\tilde{\boldsymbol{x}})$ will fit a biased distribution. On the other hand, perturbing the testing set in poisoning attack is similar to the case in evasion attack. The model is well trained on clean data and tested on a biased distribution. Our empirical results demonstrate that gradient-based methods tend to perturb the local structure of training nodes if the size of the training set is small. We speculate that it is because in SSNC, the smaller the training set, the more effective it is to attack the structure of the training nodes (**Theorem 4.1**, we give the proof in Appendix A.3).

**Assumption 4.1.** *We consider a graph $\mathcal{G}$, where each node $i$ has feature $\boldsymbol{x}_i \in \mathbb{R}^d$ and label $y_i \in \{0, 1\}$. We assume that (1) $\mathcal{G}$ is k-regular; (2) The feature of arbitrary node $i$ is sampled from the normal distribution $\mathcal{N}(\boldsymbol{\mu}_{y_i}, \Lambda)$ associated with its label and independent to other nodes. $\Lambda$ is a diagonal matrix and same for class 0 and 1; (3) The graph is a homophilous graph. The homophily ratio is h, which means each node connects $kh$ nodes with the same label, and $0.5 \le h \le 1$.*

**Theorem 4.1.** *Consider a graph $\mathcal{G}$, which follows Assumptions 4.1. The perturbation rate is $\Delta$, and the training size is t. The smaller the size of the training set, the larger the distribution shift caused by uniformly inserting heterophily edges (edges link two nodes with different labels) into the training set.*

MetaAttack's success in poisoning attack (as shown in Table 4) can be accounted for the same reason. It can adaptively adjust **IS** so that it attacks the training set when the training size is small but changes to attack the testing set as the size of the training set increases. Notably, the performance of MetaAttack shows a decreasing and then increasing trend as the training set increases, so such a strategy can effectively enlarge the distribution shift.

### 4.3 Impact of Surrogate Loss and Gradient Computation

When implementing the attack algorithms, two factors influence the distribution of adversarial edges: (1) surrogate loss and (2) how the gradient is obtained. Ma et al. (2020); Geisler et al. (2021) have studied the impact of various surrogate loss on the attack performance, such as Cross-Entropy (CE), Carlini-Wagner (CW), Non-targeted CE, *etc.* Different from them, we are concerned with the question of which nodes are used to calculate the loss. $\mathcal{L}_{self} = \mathcal{L}(f_{\theta^*}(\hat{\mathbf{A}}, \mathbf{X})_U, \hat{\boldsymbol{y}}_U)$ and $\mathcal{L}_{train} = \mathcal{L}(f_{\theta^*}(\hat{\mathbf{A}}, \mathbf{X})_L, \boldsymbol{y}_L)$ are two widely used loss. We find that they lead to different attack tendencies. Attack methods with $\mathcal{L}_{train}$ and $\mathcal{L}_{self}$ will focus on the training and testing nodes, respectively. In addition, MetaAttack is a special case due to its ability to adjust the location of adversarial edges adaptively. We find that such adaptability comes from its way of calculating the gradient, *i.e.*, meta gradient. For more details, see A.4.

## 5 Explanation of Phenomena in Graph Attacks

With the advantage of the new perspective and the formulation of distribution shift, we could introduce some other phenomena in graph attacks and explain them.

### 5.1 Tend to Insert but Not Delete

Previous work shows that gradient-based attack methods tend to insert but not delete edges Zügner & Günnemann (2019); Lin et al. (2020); Xu et al. (2019a). For example, roughly 80% of perturbations in MetaAttack are insertions, and as expected by the homophily assumption, edges inserted connect nodes from different classes in most cases. Why is insertion more destructive? We argue that insertion can cause a larger distribution shift and provide a theoretical justification (**Theorem 5.1**).

**Theorem 5.1.** *Consider a graph $\mathcal{G}$, which follows Assumptions 4.1. If $\frac{\Delta k}{t(k+1)} \le 1 - \ln 2$ and $(2h-1)t > \Delta$, inserting heterophily edges in the training set will cause a larger distribution shift than deleting homophily edges.*

We provide the proof in Appendix A.5. Since adversarial attacks aim at performing unnoticeable perturbations, the perturbation rate $\Delta$ is often very small, and in homophilous graphs, $h$ is often much larger than $0.5$ in homophilous graphs. Therefore, the assumption $\frac{\Delta k}{t(k+1)} \leq 1 - \ln 2$ and $(2h-1)t > \Delta$ could be generally satisfied. To ensure this tendency is not caused by the sparsity of the graph, we build a dense synthetic graph and attack it by MetaAttack in Table 10 in Appendix A.5. The results hold the same conclusion that attack methods tend to insert but not delete.

## 5.2 GRADIENT-BASED METHODS OUTPERFORM HOMOPHILY-BASED METHODS

Under the homophily assumption McPherson et al. (2001), *i.e.*, connected nodes are more likely to have similar features and labels; one mainstream opinion is that attack methods tend to increase the heterophily of the graph Zügner & Günnemann (2019); Wu et al. (2019); Zhu et al. (2021a). The homophily assumption ignores the location of the perturbations, so it cannot account for the superiority of the gradient-based approaches over DICE Waniek et al. (2018), a heuristic method that directly increases the heterophily of the graph by randomly connecting nodes from different classes and disconnecting nodes from the same class. However, this is not surprising from a distribution perspective.

On the one hand, DICE randomly attacks the entire graph, resulting in synchronous perturbation of $p_{train}$ and $p_{test}$, while the gradient-based methods concentrate on attacking the training set. According to Theorem 4.1, attacking the smaller part is more effective to increase the distribution shift, so DICE performs worse. On the other hand, DICE algorithm perturbs both training and testing nodes using a similar rule, wherein heterophily is increased. Consequently, $p_{train}$ and $p_{test}$ may exhibit a biased shift in the same direction, leading to a reduction in distribution shift rather than an increase. We speculate that attack methods might be more destructive if they mainly perturb one of $p_{train}$ and $p_{train}$. To support this, We turn the Train-Test perturbations generated by MetaAttack into direct edges. Fig. 3 shows the results and demonstrates two critical points. First, Directed→Train outperforms the vanilla MetaAttack, indicating that simultaneously perturbing $p_{train}$ and $p_{test}$ is worse than primarily perturb $p_{train}$, indicating that perturbing $p_{train}$ and $p_{test}$ might be biased in the same direction. Second, Direct→Test is much weaker than Direct→Train. That is to say, what makes MetaAttack so destructive is the perturbations to the smaller part, training nodes.

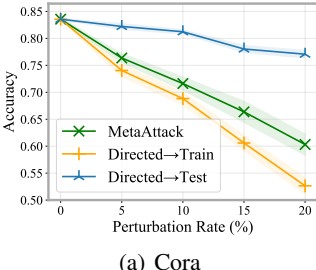
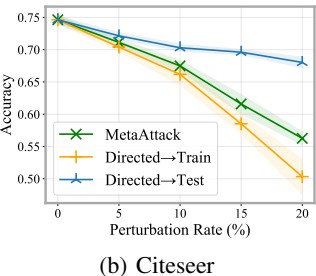

(a) Cora        (b) Citeseer

Figure 3: The performance of GCN attacked by MetaAttack and two variants. Directed→Train variant prevented testing nodes from aggregating information from training nodes through adversarial edges in Train-Test, while Directed→Test is on the contrary. The data split follows 10%/ 10%/ 80/%(train/ val/test).

## 5.3 HIGH-DEGREE NODES

High-degree nodes are special. First, attack algorithms often avoid modifying the local structure around high-degree nodes Zügner & Günnemann (2019). The more neighbors a node has, the more stable its community feature is. Attacking them will make less change to the distribution. Additionally, high-degree nodes are naturally high-density data (see A.6). High-density data possess two important properties: (1) They are easier to be classified correctly. (2) They are reliable neighbors to other nodes because they are located in high-density areas of the true distribution and are not attacked. We can leverage them to improve the robust GNNs (Table 11). For example, trust them more (for more details, see A.6).

## 6 PRACTICAL TIPS

With all the observation and analysis, we put forward several concrete tips to improve attack and defense methods.

### 6.1 POISONING ATTACK

**Tip 1: It is better to focus on attacking the smaller part.** For poisoning attack, the performance of GNNs can be degrade by perturbing $p_{train}$ or $p_{test}$. According to Theorem 4.1, the smaller the size of the data set is, the greater the distribution is changed by injecting a fixed number of perturbations. MetaAttack is a good example of this tip, in which the perturbations are generated according to the size of the training set. We improve DICE and random perturbations (Random) along these lines and compare them to the vanilla version in Fig. 4. For Cora, we focus the attack on the training nodes. As the public split on ogbn-arxiv is approximately 54%/17%/29% (train/val/test), we conduct the modifications around the testing nodes. We significantly improve the performance of both DICE and Random. The improvement is more remarkable in ogbn-arxiv. We conclude that such a strategy can be well-applied to large-scale graphs.

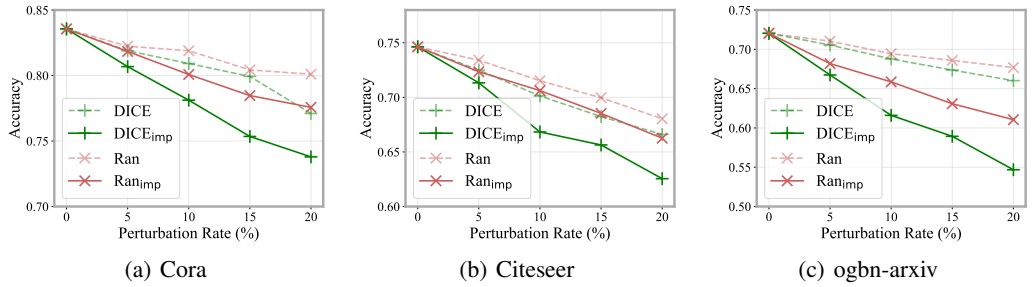

|     |     |     |
| --- | --- | --- |
| (a) Cora | (b) Citeseer | (c) ogbn-arxiv |

Figure 4: Attacking GCN on Cora, Citeseer and ogbn-arxiv where the $\mathbf{DICE_{imp}}$ and $\mathbf{Ran_{imp}}$ denote variants of DICE and Random that attack the smaller part of the graph.

**Tip 2: Meta gradient is powerful.** According to the comparison between MetaAttack and other variants, we conclude that meta gradient is a convenient tool in poisoning attack. Regardless of how the data set is split, meta gradient can help the model adaptively adjust the distribution of perturbations to enlarge the distribution shift effectively.

**Tip 3: Directed attack can outperform undirected attack.** The results of Fig. 3 indicate that an adversarial edge can affect the aggregation of two end nodes, but only one of them may contribute to the accuracy decrease. It also suggests that perturbing both $p_{train}$ and $p_{test}$ may not be a good idea as $p_{train}$ and $p_{test}$ might be skewed in the same direction, failing to increase the distribution shift and, consequently, failing the attack.

### 6.1.1 A HEURISTIC ATTACK ALGORITHM

Considering all the properties of the gradient-based attacks from the data distribution perspective, we propose a simple heuristic poisoning attack algorithm that can achieve comparable performance and scale to large graphs. We provide the overall training algorithm, performance, and the runtime comparisons in A.7. Its core idea is to perturb the more easily perturbed distribution in $p_{train}$ and $p_{test}$ to enlarge the distribution shift.

### 6.2 EVASION ATTACK

**Tip 4: Do not waste the limited bullets on the training nodes.** The classifier is already trained in evasion attack. In other words, $p_\theta(y|\tilde{x})$ is fixed, while the only effect of attacking the training set is to bias the model. Thus, attackers should avoid modifying the structures of training nodes, especially the edges in the Train-Train area. According to Table 5 and Table 7, $\mathcal{L}_{self}$ should be eschewed when meta-gradients are not used.

**Tip 5: The evasion attack is unlikely to cause higher performance degradation than the poisoning.** Essentially, evasion attacks have a smaller effective range than poisoning attacks. Attackers can perform poisoning attack in an evasion form. As we mentioned before, only modifications around the testing nodes can degrade the performance, so one can make exactly the same perturbations to the test nodes when carrying out the poisoning attack. In this way, the model will be trained on nearly clean data and tested on contaminated data. In short, poisoning attack can at least completely replicate evasion attack.

## 6.3 Defense

**Tip 6: Defend the vulnerable part of the graph.** To defend against graph adversarial attacks, many studies have been proposed around the central concept of Graph Structure Learning (GSL) Li et al. (2022); Wu et al. (2019); Zhang & Zitnik (2020); Jin et al. (2020); Zhu et al. (2021b), which aims to optimize the perturbed structure. An attacker can achieve good results by attacking distributions that can be easily perturbed. Likewise, the defender can mainly optimize the corresponding structures. Jaccard and GNNGuard Zhang & Zitnik (2020) are two representative GSL methods that refine the graph structure via the similarity of the node features. In Table 2, we present the experimental results for vanilla Jaccard and GNNGuard and the results if they only refine the structure of the training set. The improvement implies that the modifications made by vanilla Jaccard and GNNGuard on the testing structure are helpless.

**Tip 7: The high-degree nodes can help a lot.** If the true distribution is like the normal distribution with the highest probability density located around the mean, the mean aggregation can reduce the variance of the center nodes. That is to say, high-degree nodes are high-density data and trustworthy. We provide an example of leveraging them to enhance the robustness of GNN in Table 11.

**Tip 8: We can improve the robustness by decreasing the distribution shift.** Once we know that the effectiveness of the attack algorithm stems from increasing the distribution shift, we can enhance the robustness by directly eliminating the inconsistency between the training set and

Table 2: Adversarial accuracy(%) on Cora and Citeseer attacked by MetaAttack (stronger is bold). The asterisk indicates that the GSL methods only optimize the local structure of the traning nodes.

| Datasets | Ptb rate | GCN | Jaccard(*) | GNNGuard(*) |
|---|---|---|---|---|
| Cora | 0% | 83.56 | 81.79(**83.01**) | 78.52(**82.59**) |
| | 5% | 76.36 | 80.23(**82.03**) | 77.96(**81.17**) |
| | 10% | 71.62 | 74.65(**77.51**) | 74.86(**78.53**) |
| | 15% | 66.37 | 74.29(**76.51**) | 74.15(**77.21**) |
| | 20% | 60.31 | 73.11(**74.60**) | 72.03(**75.88**) |
| Citeseer | 0% | 74.63 | 73.64(**74.58**) | 70.07(**74.28**) |
| | 5% | 71.13 | 71.15(**72.23**) | 69.43(**72.34**) |
| | 10% | 67.49 | 69.85(**70.65**) | 67.89(**71.70**) |
| | 15% | 61.59 | 67.50(**68.86**) | 69.14(**70.59**) |
| | 20% | 56.26 | 67.01(**67.66**) | 69.20(**70.36**) |

the testing set. Based on this, we design a robust GNN STRG in Appendix A.8, which outperforms the SOTA methods. STRG leverages the local structures of testing nodes and pseudo-labels to train a GCN; in this case, $p_{train}$ and $p_{test}$ can be regarded as almost the same.

## 6.4 For Datasets

**Tip 9: Data split is the non-negligible part of graph adversarial attack**. It is hard to maliciously manipulate the prediction made by GNNs without the data split. Meanwhile, the data split significantly affects the evaluation of attack and defense methods' effectiveness. We provide the details in A.9.

## 7 Conclusion

To better understand attacks on graphs, we revisit graph adversarial attack from a data distribution perspective and formulate the distribution shift in SSNC. Based on this, we argue that the tendencies of gradient-based methods and their destructive power essentially comes from increasing the distribution shift. We put forward several practical tips underpinned by what we found and provide some uses of the tips. Additionally, we give some open research ideas and hope they can spur research in this area.

## 8 REPRODUCIBILITY STATEMENT

The two proposed algorithms are not our focus in this work, and the key to their success is the thinkings behind them but not a technical novelty. They are both *easy but effective*. All the details are mentioned in the pseudo-code in Algorithm 1 and Algorithm 2, and the codes are provided at https://github.com/likuanppd/STRG. For other baselines, we provide the implementation details in Appendix A.1.

## 9 ETHIC STATEMENT

The robustness of GNNs has become an emerging research problem, especially for security-critical domains, *e.g.*, credit scoring or fraud detection. For instance, in graph fraud transaction detection, fraudsters can conceal themselves by deliberately dealing with common users, which may generate adversarial edges. In this work, we provide a new perspective on studying the robustness of graph models. A better understanding of the attack methods and the structural vulnerability of GNNs can help us improve the security level in these domains. Meanwhile, although we offer examples of how to leverage the tips, most of them can be further explored and used in more sophisticated ways. In conclusion, we think this work will not pose a security risk and can positively affect research in this area.

## 10 ACKNOWLEDGEMENT

The research work supported by National Key R&D Plan No. 2022YFC3303302, the National Natural Science Foundation of China under Grant No. 61976204. Xiang Ao is also supported by the Project of Youth Innovation Promotion Association CAS and Beijing Nova Program Z201100006820062.

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

## A APPENDIX

### A.1 IMPLEMENTATION DETAILS

We conduct our experiments on four benchmark datasets including Cora, Citeseer Sen et al. (2008), Polblogs Adamic & Glance (2005), and one large-scale citation graph ogbn-arxiv Hu et al. (2020). All the experiments For Cora, Citeseer, and Polblogs, the data split follows 10%/10%/80% if not specified. For the ogbn-arxiv we use the public split. Following Zügner et al. (2018); Jin et al. (2020), for Cora, Polblogs, and Citeseer, we only consider the largest connected component (LCC). The statistics are listed in Table 3. There are no features available in Polblogs. Following Jin et al. (2020); Liu et al. (2021a) we set the feature matrix to be a $n \times n$ identity matrix.

Table 3: Dataset statistics.

| Datasets | $N_{LCC}$ | $E_{LCC}$ | Classes | Features |
|---|---|---|---|---|
| Cora | 2,485 | 5,069 | 7 | 1,433 |
| Citeseer | 2,110 | 3,668 | 6 | 3,703 |
| Polblogs | 1,222 | 16,714 | 2 | / |
| ogbn-arxiv | 169,343 | 1,166,243 | 40 | 128 |

We use DeepRobust, an adversarial attack repository Li et al. (2020), to implement MetaAttack Zügner & Günnemann (2019), PGD Xu et al. (2019a), DICE Waniek et al. (2018), Jaccard Wu et al. (2019), SimpGCN Jin et al. (2021) and ProGNN Jin et al. (2020). We perform FGSM according to Dai et al. (2018). STABLE Li et al. (2022), GNNGuard, and Elastic Liu et al. (2021a) are implemented with the code provided by the authors.

All the hyper-parameters are tuned based on the loss and accuracy of the validation set. For Jaccard, the Jaccard Similarity threshold are tuned from {0.01, 0.02, 0.03, 0.04, 0.05}. For GNNGuard, ProGNN, SimpGCN, and Elastic, we use the default hyper-parameter settings in the authors' implementation.

## A.2 THE DISTRUBION OF ADVERSARIAL EDGES

Table 4 shows the distribution of adversarial edges on the graph, attacked by MetaAttack, PGD, and FGSM Dai et al. (2018). Only MetaAttack can adjust the distribution of the perturbations according to the size of the training set. Specifically, it consistently generates perturbations around the smaller one in the training set and the testing set. PGD and FGSM always focus on attacking training nodes, resulting in poor performance when the training set size becomes large.

Fig. 5 is the visualization of the adjacency matrices of Cora and Citeseer. Compared with the results in Table 4 and Table 1, we can see that **IS** can reflect the density of perturbations in each area of the adjacency matrix. MetaAttack adjust such density according to the size of training set. It focuses on attacking the smaller of the training set and the testing set, and the perturbations are nearly uniformly distributed when the training size is 0.5. On the contrary, PGD and FGSM consistently attacks the training nodes regardless of the data split. Consequently, MetaAttack can easily fool GCN no matter how large is the training set, while PGD failed when the training size increases.

Zhu et al. (2021a) shows that MetaAttack fails on evasion attack with a 10%/10%/80% data split, and it is consistent with our discovery. When the training set is small, such adaptivity will make MetaAttack focus on modifying the local structure around the training set. In the setting of evasion attack, the model is already trained on unbiased data, so testing on unchanged data does not cause worse performance. For example, when the training size is 10%, $\lambda_3 \approx 0.9 \times 0.9 = 0.81$ and $IS_3 = 0.007$, meaning that the Test-Test is the largest part of the graph, but most of the local structure around testing nodes is not malicious modified.

Table 4: The **IS** of MetaAttack, PGD, and FGSM under 10% perturbation rate. The Acc indicates the adversarial accuracy, and for attack methods a lower value is better. Clean is the accuracy of the vanilla GCN on clean graphs. We highlight the strongest attack in bold.

| Dataset | Training set | MetaAttack Train-Train $(IS_1)$ | Train-Test $(IS_2)$ | Test-Test $(IS_3)$ | Acc | PGD Train-Train $(IS_1)$ | Train-Test $(IS_2)$ | Test-Test $(IS_3)$ | Acc | FGSM Train-Train $(IS_1)$ | Train-Test $(IS_2)$ | Test-Test $(IS_3)$ | Acc | Clean |
|---|---|---|---|---|---|---|---|---|---|---|---|---|---|---|
| | 10% | 11.053 | 4.935 | 0.007 | **67.49** | 17.650 | 4.601 | 0.000 | 69.56 | 22.131 | 4.311 | 0.000 | 69.99 | 74.63 |
| | 20% | 3.620 | 2.655 | 0.009 | **72.27** | 8.562 | 2.055 | 0.000 | 73.38 | 10.041 | 1.853 | 0.000 | 71.23 | 74.73 |
| | 30% | 2.884 | 1.646 | 0.100 | **75.81** | 5.142 | 1.279 | 0.000 | 76.15 | 5.556 | 1.190 | 0.000 | 76.12 | 76.26 |
| | 40% | 1.554 | 1.332 | 0.311 | **75.95** | 3.666 | 0.861 | 0.000 | 77.20 | 3.859 | 0.791 | 0.000 | 77.15 | 77.19 |
| Citeseer | 50% | 1.145 | 1.175 | 0.514 | **75.71** | 2.785 | 0.602 | 0.000 | 77.59 | 2.962 | 0.503 | 0.000 | 77.61 | 77.66 |
| | 60% | 1.093 | 1.175 | 0.717 | **77.88** | 2.165 | 0.460 | 0.000 | 79.11 | 2.072 | 0.541 | 0.000 | 78.84 | 79.26 |
| | 70% | 0.809 | 1.177 | 1.213 | **75.64** | 1.744 | 0.347 | 0.000 | 77.73 | 1.795 | 0.286 | 0.000 | 77.56 | 77.68 |
| | 80% | 0.481 | 1.868 | 2.414 | **76.63** | 1.424 | 0.271 | 0.000 | 79.14 | 1.473 | 0.171 | 0.000 | 79.15 | 79.14 |
| | 90% | 0.334 | 3.749 | 5.142 | **76.51** | 1.181 | 0.245 | 0.000 | 82.22 | 1.181 | 0.197 | 0.000 | 82.14 | 82.26 |
| | 10% | 9.095 | 4.394 | 0.156 | **70.4** | 26.708 | 3.918 | 0.048 | 80.85 | 28.606 | 3.956 | 0.000 | 82.56 | 95.04 |
| | 20% | 3.122 | 2.228 | 0.255 | **74.17** | 8.651 | 2.029 | 0.010 | 85.64 | 9.141 | 1.975 | 0.000 | 86.66 | 95.33 |
| | 30% | 1.494 | 1.475 | 0.504 | **73.60** | 4.239 | 1.467 | 0.009 | 86.43 | 4.688 | 1.365 | 0.001 | 87.22 | 95.21 |
| | 40% | 1.028 | 1.166 | 0.766 | **73.44** | 3.065 | 1.054 | 0.015 | 88.85 | 3.351 | 0.964 | 0.000 | 89.48 | 95.49 |
| Polblogs | 50% | 0.776 | 1.263 | 0.971 | **73.95** | 2.192 | 0.901 | 0.002 | 88.88 | 2.387 | 0.807 | 0.000 | 90.51 | 95.36 |
| | 60% | 0.642 | 1.183 | 1.251 | **76.19** | 1.733 | 0.787 | 0.000 | 89.18 | 1.837 | 0.701 | 0.000 | 91.57 | 95.97 |
| | 70% | 0.399 | 1.440 | 2.191 | **73.35** | 1.418 | 0.730 | 0.000 | 90.37 | 1.477 | 0.654 | 0.000 | 91.33 | 95.67 |
| | 80% | 0.260 | 2.066 | 4.194 | **72.50** | 1.225 | 0.680 | 0.000 | 90.00 | 1.327 | 0.443 | 0.000 | 92.01 | 96.85 |
| | 90% | 0.299 | 3.734 | 8.558 | **76.13** | 1.140 | 0.435 | 0.000 | 92.76 | 1.124 | 0.492 | 0.000 | 94.83 | 98.32 |

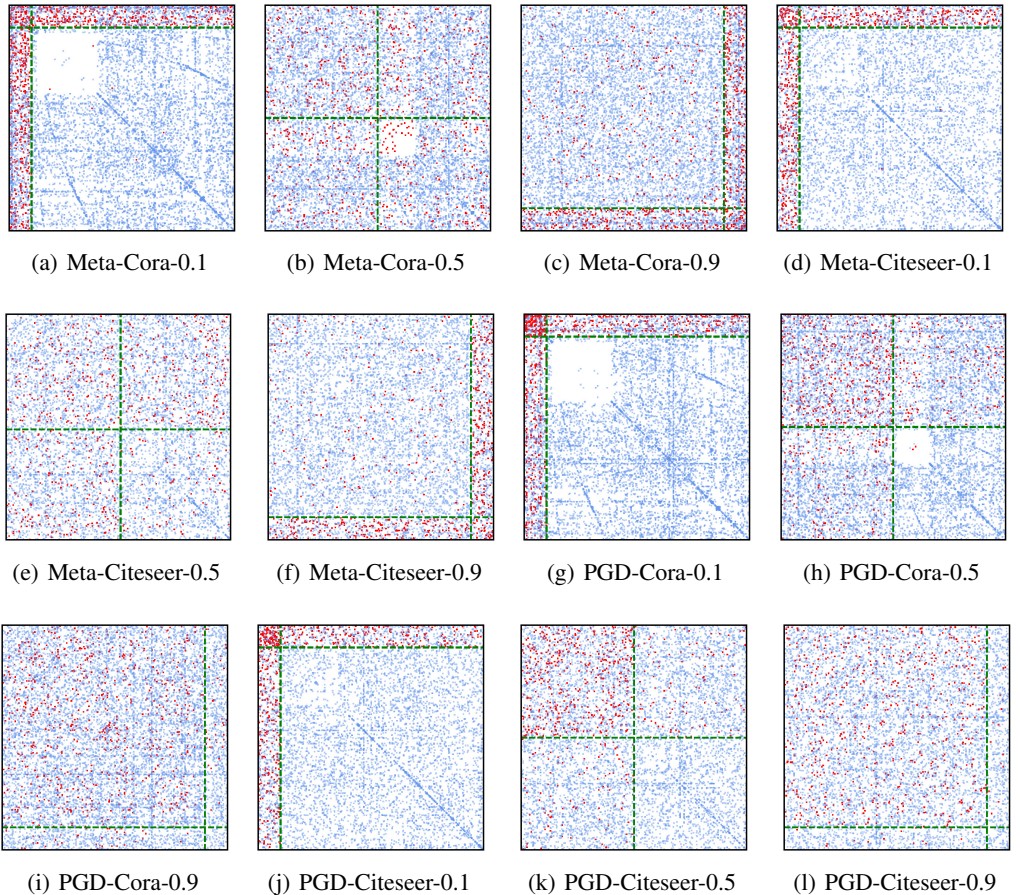

Figure 5: The adjacency matrix of Cora and Citeseer attacked by MetaAttack and PGD with different training size.

### A.3 PROOF OF THEOREM 4.1

*Proof.* There are $N$ nodes and $\frac{Nk}{2}$ edges in graph $\mathcal{G}$. Totally $\frac{\Delta Nk}{2}$ will be inserted in the graph. Each node in the training set will have $2\frac{\frac{\Delta Nk}{2}}{Nt} = \frac{\Delta k}{t}$ more neighbours with different label due to the uniform insertions. Thus, the community features of the nodes in class 0 can be viewed as sampled from a biased distribution $p_{train}(\tilde{\boldsymbol{x}}|y=0)$:

$$\mathcal{N}(\frac{(1+hk)\boldsymbol{\mu}_0 + (k - hk + \frac{\Delta k}{t})\boldsymbol{\mu}_1}{k + 1 + \frac{\Delta k}{t}}, \frac{\Lambda}{k + 1 + \frac{\Delta k}{t}}). \tag{7}$$

The structures of testing nodes are not modified, so $p_{test}(\tilde{\boldsymbol{x}}|y=0)$ is:

$$\mathcal{N}(\frac{(1+hk)\boldsymbol{\mu}_0 + (k - hk)\boldsymbol{\mu}_1}{k + 1}, \frac{\Lambda}{k + 1}). \tag{8}$$

Let $\boldsymbol{\delta}_0$ and $\boldsymbol{\delta}_1$ denote the mean of $p_{train}(\tilde{\boldsymbol{x}}|y=0)$ and $p_{test}(\tilde{\boldsymbol{x}}|y=0)$, respectively. According to Eq. (6) and the KL-Divergence formula for two normal distribution, we have

$$D_{KL}(p_{train}(\tilde{\boldsymbol{x}}|y=0),\ p_{test}(\tilde{\boldsymbol{x}}|y=0))$$
$$= \frac{1}{2}\Big\{\log\frac{(k + 1 + \frac{\Delta k}{t})^d}{(k+1)^d} + \frac{d(k+1)}{k + 1 + \frac{\Delta k}{t}} +$$
$$(k+1)(\boldsymbol{\delta}_0 - \boldsymbol{\delta}_1)^T \Lambda^{-1}(\boldsymbol{\delta}_0 - \boldsymbol{\delta}_1) - d\Big\} \tag{9}$$

Let $\frac{k+1+\frac{\Delta k}{t}}{k+1} = S$, then the first two terms of Eq. (9) can be rewritten as

$$\log S^d + \frac{d}{S}, \quad S > 1. \tag{10}$$

Take the derivative of Eq. (10) w.r.t. to $S$

$$\frac{d}{S}\left(\frac{1}{\ln 2} - \frac{1}{S}\right), \tag{11}$$

apparently it is larger than 0 for $S > 1$, so Eq. (10) is monotonically increasing w.r.t $S$ and monotonically decreasing w.r.t. the size of training set $t$.

For the third term of Eq. (9), we first calculate $\boldsymbol{\delta}_0 - \boldsymbol{\delta}_1$. For notational simplicity, we let $P = (1 + hk)\boldsymbol{\mu}_0 + (k - hk)\boldsymbol{\mu}_1$

$$\begin{aligned}
\boldsymbol{\delta}_0 - \boldsymbol{\delta}_1 &= \frac{P + \frac{\Delta k}{t}\boldsymbol{\mu}_1}{k + 1 + \frac{\Delta k}{t}} - \frac{P}{k+1} \\
&= \Delta k \frac{(k+1)\boldsymbol{\mu}_1 - P}{(tk + t + \Delta k)(k+1)}.
\end{aligned} \tag{12}$$

Then we let $\Delta k \frac{(k+1)\boldsymbol{\mu}_1 - P}{(tk+t+\Delta k)(k+1)} = v$. Apparently, $v^T v$ is monotonically decreasing w.r.t. $t$, and we can easily prove $v^T \Lambda^{-1} v$ is also monotonically decreasing, because $\Lambda^{-1}$ just introduces constants into each element of $v$.

Therefore, Eq. (9) will become larger as $t$ gets smaller. This also holds for nodes in class 1. Then we can get the conclusion that the smaller the $t$, the larger the distribution shift. $\qquad\square$

### A.4 IMPACT OF SURROGATE LOSS AND GRADIENT COMPUTATION

**The Impact of $\mathcal{L}_{train}$ or $\mathcal{L}_{self}$**  Table 5 shows the attack tendencies and performance of PGD with $\mathcal{L}_{train}$ or $\mathcal{L}_{self}$. We only provide the results on Cora, but similar results are observed on other commonly used datasets for SSNC. $\mathcal{L}_{train}$ will make the attack algorithm tend to attack the training nodes, while $\mathcal{L}_{self}$ will make it attack the testing nodes.

Table 5: The location statistics of adversarial edges on Cora under 10% perturbation. Num represents the number of adversarial edges in the corresponding area. Evasion and Poison are the adversarial accuracy of GCN Kipf & Welling (2017) under evasion and poisoning attack. We highlight the stronger attack in bold.

| Attack | IS$_1$/Num | IS$_2$/Num | IS$_3$/Num | clean | Evasion | Poison |
|---|---|---|---|---|---|---|
| PGD$_{train}$ | 22.18/224 | 4.28/778 | 0.00/2 | 83.56 | 83.39 | **77.92** |
| PGD$_{self}$ | 0.80/8 | 1.40/252 | 0.91/740 | 83.56 | **81.94** | 83.55 |

$\mathcal{L}_{train}$ focuses on the loss of the training nodes, which is mainly related to the local structure of these nodes. Therefore, when computing the gradient $\nabla_{\mathbf{A}} \mathcal{L}_{atk}$, only the part of the adjacency matrix associated with the training nodes will be modified, *i.e.*, the **Train-Train** and **Train-Test**. PGD$_{train}$ works in poisoning attack but fails in evasion attack. This result is as expected because $p_{train}$ is well-perturbed but $p_{test}$ nearly remains unchanged. There are some edges in the Train-Test, but the largest part of the graph, the Test-Test, is barely changed.

$\mathcal{L}_{self}$ utilizes the pseudo labels of the unlabeled nodes to compute the loss, so the majority of modifications are around the unlabeled nodes. Although PGD$_{self}$ seems to work in evasion attack, the degradation in accuracy is not as pronounced as PGD$_{train}$ in poisoning attack. We suggest that there are mainly two reasons: (1) The testing set is much larger than the training set, so the same perturbation brings a smaller change in $p_{test}$ than in $p_{train}$. (2) We use Cross-Entropy Loss to conduct these experiments. Geisler et al. (2021) shows that CE Loss makes algorithms primarily attack nodes that are already misclassified. Such attacks are valid in poisoning attack and can further bias the model $p_\theta(y|\tilde{\boldsymbol{x}})$. However, in evasion attack, attacking misclassified nodes will not bring any

drop in accuracy.Another example that uses $\mathcal{L}_{self}$ is PR-BCD Geisler et al. (2021), which is an novel evasion attack. We present the location statistics of perturbations on Citeseer attacked by it in Table 6. We can see that all the perturbations are generated around the testing nodes, leading to effective performance.

Table 6: The location statistics of adversarial edges on Citeseer attacked by PR-BCD.

| Ptb Rate | Train-Train | Train-Test | Test-Test |
|---|---|---|---|
| 5% | 0 | 15 | 168 |
| 10% | 0 | 29 | 322 |
| 15% | 0 | 55 | 488 |
| 20% | 0 | 69 | 665 |

**Gradient Computation** MetaAttack is a special case due to its ability to adjust the distribution of adversarial edges. Even today, many practitioners still consider it to be the SOTA of poisoning attacks on small-scale graphs. According to the Theorem 4.1, this adaptivity makes MetaAttack incline to attack the more easily perturbed distributions in $p_{train}$ and $p_{test}$. In other words, it identifies structural modifications that significantly increase distribution shift.

We find that this adaptability comes from its way of calculating the gradient. Let's review the meta-gradient expressed in Zügner & Günnemann (2019):

$$\nabla_{\mathbf{A}}\mathcal{L}_{atk}(f_{\theta_T}(\mathbf{A},\mathbf{X})) = \nabla_f \mathcal{L}_{atk}(f_{\theta_T}(\mathbf{A},\mathbf{X})) \cdot [\nabla_{\mathbf{A}} f_{\theta_T}(\mathbf{A},\mathbf{X}) + \nabla_{\theta_T} f_{\theta_T}(\mathbf{A},\mathbf{X}) \cdot \nabla_{\mathbf{A}}\theta_T],$$
(13)

$$\nabla_{\mathbf{A}}\theta_{t+1} = \nabla_{\mathbf{A}}\theta_t - \alpha\nabla_{\mathbf{A}}\nabla_{\theta_t}\mathcal{L}_{train}(f_{\theta_t}(\mathbf{A},\mathbf{X}) \tag{14}$$

where $T$ is the training steps to obtain the optimal parameter via vanilla gradient decent with learning rate $\alpha$. The parameters $\theta$ are often fixed and detached when calculating the gradient $\nabla_{\mathbf{A}}\mathcal{L}_{atk}$ in other gradient-based methods Xu et al. (2019a); Wu et al. (2019). In MetaAttack, however, $\theta$ is iteratively retrained as the graph is gradually contaminated, and the derivatives w.r.t. the adjacency matrix are taken into account.

**The effectiveness and adaptive capability of MetaAttack stems from this gradient calculation method.** MetaAttack will perform similar to PGD if the parameters are *fixed or detached from the gradient computation*. Table 7 shows the performance of $\text{Meta}_{self}$, $\text{Meta}_{train}$, $\text{Detach}_{self}$, $\text{Detach}_{train}$, $\text{Fix}_{self}$, and $\text{Detach}_{train}$. Detach and Fix are two variants of MetaAttack. The former detaches the parameters from the meta gradient calculation and the latter fixes the parameters after training in the clean graph. $\text{Meta}_{self}$ is the only one that always attacks the more easily perturbed distribution in $p_{train}$ and $p_{test}$. Other attack methods with $\mathcal{L}_{self}$ will only attack the testing nodes, but $\text{Meta}_{self}$ will attack the training nodes when the size of the testing set is samll. It is because the meta gradient will take the training process into account by Eq. 3, which is associated with the training nodes. Thus, the local structure of the training nodes is related to the meta gradient computation. Detach and Fix perform like PGD that the tendency is closely related to the surrogate loss, *i.e.*, $\mathcal{L}_{self}$ or $\mathcal{L}_{train}$.

As expected, $\text{Meta}_{self}$ outperforms all the variants regardless of the data split. It is worth noting that variants with $\mathcal{L}_{train}$ usually performs better when the training set is small, while variants with $\mathcal{L}_{self}$ are more destructive when the testing set is small.

**Different Surrogate Loss** To ensure this tendency is not caused by a specific type of loss, we explore the effect of other two losses in Graph Attack, including the Carlini-Wagner loss Xu et al. (2019a) CW= $\min(\max_{c \neq c^*} z_c - z_{c^*}, 0)$.and Masked Cross Entropy MCE Geisler et al. (2021) $= 1/\mathbb{V}^+ \mid \sum_{i \in \mathbb{V}+} -\log(p_{c^*}^{(i)})$, where $c^*$ is the ground truth label and $\mathbb{V}^+$ indicates correctly classified nodes. For CW and MCE, we can also divide them into $\mathcal{L}_{self}$ and $\mathcal{L}_{train}$ according to which nodes are used to calculate the loss. In Table 8, we present the **IS** for $\text{Meta}_{self}$ with CW loss and MCE loss. $\text{Meta}_{self}$ with CW and MCE can also adjust the distribution of the perturbations according to the size of the training set, which is consistent with CE loss. With all the results, we conclude that

Table 7: The location statistics of perturbations and the performance of GCN poisoning attack on Cora under 10% perturbation. $IS_3$-0.8 is the $IS_3$ when the size of the training set is 0.8 and reveals the adaptivity of different variants. We highlight the strongest attack method in bold.

| Attack | $IS_1$/Num | $IS_2$/Num | $IS_3$/Num | Poison | $IS_3$-0.8 | Poison-0.8 |
|---|---|---|---|---|---|---|
| $Meta_{train}$ | 14.46/73 | 4.75/432 | 0.00/1 | 73.37 | 0.00 | 88.35 |
| $Meta_{self}$ | 11.88/60 | 4.86/442 | 0.01/3 | **71.62** | 3.409 | **80.32** |
| $Detach_{train}$ | 14.65/74 | 4.74/431 | 0.00/1 | 79.94 | 0.00 | 87.52 |
| $Detach_{self}$ | 0.00/0 | 0.80/73 | 1.06/433 | 81.73 | 23.56 | 81.44 |
| $Fix_{train}$ | 15.05/76 | 4.64/422 | 0.20/8 | 79.63 | 0.00 | 89.56 |
| $Fix_{self}$ | 0.00/0 | 0.85/77 | 1.05/429 | 82.80 | 26.34 | 83.53 |

the key to affecting the distribution of adversarial edges is which nodes are used to calculate the loss rather than the type of surrogate loss.

Table 8: The **IS** of $Meta_{self}$ with two different surrogate losses, CW, and MCE under 10% perturbation rate. The Acc indicates the adversarial accuracy, and for attack methods a lower value is better. Clean is the accuracy of the vanilla GCN on clean graphs.

| Dataset | Training set | CW Train-Train ($IS_1$) | Train-Test ($IS_2$) | Test-Test ($IS_3$) | Acc | MCE Train-Train ($IS_1$) | Train-Test ($IS_2$) | Test-Test ($IS_3$) | Acc | Clean |
|---|---|---|---|---|---|---|---|---|---|---|
| Cora | 10% | 7.615 | 5.078 | 0.005 | 71.55 | 9.012 | 4.854 | 0.051 | 74.42 | 83.56 |
| | 20% | 2.953 | 2.559 | 0.096 | 82.07 | 3.969 | 2.548 | 0.043 | 78.80 | 85.67 |
| | 30% | 1.903 | 1.626 | 0.295 | 80.64 | 2.450 | 1.622 | 0.198 | 84.49 | 86.99 |
| | 40% | 1.430 | 1.231 | 0.500 | 80.07 | 1.689 | 1.132 | 0.517 | 81.88 | 87.01 |
| | 50% | 0.986 | 1.079 | 0.856 | 83.11 | 0.828 | 1.115 | 0.943 | 80.24 | 87.22 |
| | 60% | 0.625 | 1.182 | 1.300 | 83.40 | 0.718 | 1.091 | 1.361 | 83.05 | 88.04 |
| | 70% | 0.447 | 1.489 | 1.742 | 81.17 | 0.540 | 1.338 | 1.940 | 84.12 | 89.98 |
| | 80% | 0.370 | 1.942 | 3.572 | 79.72 | 0.410 | 1.979 | 2.629 | 83.52 | 88.87 |
| | 90% | 0.329 | 3.632 | 8.402 | 81.80 | 0.302 | 3.830 | 7.001 | 82.54 | 93.69 |
| Citeseer | 10% | 6.971 | 5.103 | 0.003 | 65.09 | 9.652 | 4.967 | 0.000 | 68.14 | 74.63 |
| | 20% | 3.924 | 2.569 | 0.026 | 74.03 | 3.045 | 2.646 | 0.043 | 70.77 | 74.73 |
| | 30% | 2.413 | 1.760 | 0.084 | 74.58 | 1.720 | 1.942 | 0.056 | 72.77 | 76.26 |
| | 40% | 1.598 | 1.433 | 0.152 | 75.28 | 1.411 | 1.439 | 0.228 | 72.49 | 77.19 |
| | 50% | 1.350 | 1.087 | 0.472 | 75.62 | 1.317 | 1.082 | 0.516 | 73.25 | 77.66 |
| | 60% | 0.825 | 1.208 | 0.772 | 76.31 | 0.862 | 1.134 | 0.909 | 75.13 | 79.26 |
| | 70% | 0.634 | 1.395 | 1.161 | 72.79 | 0.656 | 1.323 | 1.375 | 75.17 | 77.68 |
| | 80% | 0.409 | 1.937 | 3.034 | 73.11 | 0.520 | 1.825 | 2.138 | 72.78 | 79.14 |
| | 90% | 0.381 | 3.536 | 5.792 | 70.00 | 0.445 | 3.338 | 4.137 | 77.79 | 82.26 |
| Polblogs | 10% | 6.202 | 4.386 | 0.177 | 70.44 | 9.274 | 4.307 | 0.156 | 68.09 | 95.04 |
| | 20% | 3.409 | 2.150 | 0.272 | 73.66 | 2.948 | 2.257 | 0.247 | 69.63 | 95.33 |
| | 30% | 2.163 | 1.444 | 0.405 | 74.23 | 1.699 | 1.512 | 0.432 | 70.38 | 95.21 |
| | 40% | 1.170 | 1.214 | 0.639 | 76.79 | 1.039 | 1.214 | 0.697 | 73.80 | 95.49 |
| | 50% | 0.876 | 1.130 | 0.864 | 73.30 | 0.790 | 1.099 | 1.013 | 74.32 | 95.36 |
| | 60% | 0.527 | 1.209 | 1.435 | 73.39 | 0.574 | 1.240 | 1.237 | 74.18 | 95.97 |
| | 70% | 0.399 | 1.488 | 1.990 | 71.23 | 0.368 | 1.452 | 2.322 | 69.71 | 95.67 |
| | 80% | 0.328 | 2.144 | 2.627 | 71.74 | 0.325 | 2.110 | 2.942 | 67.52 | 96.85 |
| | 90% | 0.299 | 3.869 | 6.244 | 71.17 | 0.299 | 3.742 | 8.526 | 68.83 | 98.32 |

**Pseudo-labels in $\mathcal{L}_{self}$** Attacks with pseudo labels behave similarly to those with ground-truth labels because (see the Table 9; Meta-true means Meta-self uses the ground-truth labels to generate perturbations instead of pseudo labels). We guess it is because, in homophilous graphs, the pseudo labels are generally accurate. Thus, attacks with $\mathcal{L}_{self}$ will focus on modifying the local structure of the testing nodes to make the predictions away from the ground-truth labels. In many cases, the size of the testing set is relatively large, so it is hard to increase the distribution shift by modifying the testing structure according to Theorem 4.1. This way, methods like PGD$_{self}$ fail.

Table 9: Adversarial accuracy(%) on Cora and Citeseer attacked by Meta$_{self}$ and Meta$_{true}$.

| Datasets | Ptb rate | Meta$_{self}$ | Meta$_{true}$ |
|---|---|---|---|
| Cora | 5% | 76.36 | 75.84 |
| | 10% | 71.62 | 71.33 |
| | 15% | 66.37 | 64.42 |
| | 20% | 60.31 | 58.81 |
| Citeseer | 5% | 71.13 | 71.20 |
| | 10% | 67.49 | 67.24 |
| | 15% | 61.59 | 60.88 |
| | 20% | 56.26 | 56.14 |

## A.5 INSERTION VS. DELETION

**Proof of Theorem 5.1**

*Proof.* The Kl-Divergence between $p_{train}(\tilde{\boldsymbol{x}}|y=0)$ and $p_{test}(\tilde{\boldsymbol{x}}|y=0)$ of insertion is shown in Eq. (9), and we denote it as $D_{KL-INS}$. Similarly, we have $D_{KL-DEL}$ as followed:

$$
D_{KL-DEL}(p_{train}(\tilde{\boldsymbol{x}}|y=0),\ p_{test}(\tilde{\boldsymbol{x}}|y=0))
$$
$$
= \frac{1}{2}\Big\{ \log \frac{(k+1-\frac{\Delta k}{t})^d}{(k+1)^d} + \frac{d(k+1)}{k+1-\frac{\Delta k}{t}} + \quad (k+1)(\boldsymbol{\delta}_0' - \boldsymbol{\delta}_1)^T \Lambda^{-1}(\boldsymbol{\delta}_0' - \boldsymbol{\delta}_1) - d \Big\} \quad (15)
$$

,

where $\boldsymbol{\delta}_0' = \frac{(1+hk-\frac{\Delta k}{t})\boldsymbol{\mu}_0 + (k-hk)\boldsymbol{\mu}_1}{k+1-\frac{\Delta k}{t}}$. Now we compare $D_{KL-INS}$ and $D_{KL-DEL}$. $\frac{k+1-\frac{\Delta k}{t}}{k+1} \geq \ln 2$ due to the assumption $\frac{\Delta k}{t(k+1)} \leq 1 - \ln 2$. We already know that $\log S^d + \frac{d}{S}$ is monotonically increasing w.r.t. $S$ if $S \geq \ln 2$. Therefore,

$$
\log \frac{(k+1+\frac{\Delta k}{t})^d}{(k+1)^d} + \frac{d(k+1)}{k+1+\frac{\Delta k}{t}} > \log \frac{(k+1-\frac{\Delta k}{t})^d}{(k+1)^d} + \frac{d(k+1)}{k+1-\frac{\Delta k}{t}} \quad (16)
$$

Then, if $v^T v > (\boldsymbol{\delta}_0' - \boldsymbol{\delta}_1)^T (\boldsymbol{\delta}_0' - \boldsymbol{\delta}_1)$, we can conclude that $D_{KL-INS} > D_{KL-DEL}$.

$$
\boldsymbol{\delta}_0' - \boldsymbol{\delta}_1 = \frac{\Delta}{k} \frac{P - (k+1)\boldsymbol{\mu}_0}{(tk+t-\Delta k)(k+1)} \quad (17)
$$

Let $\boldsymbol{\delta}_0' - \boldsymbol{\delta}_1 = u$. This is equivalent to comparing $|v|$ and $|u|$.

$$
|v| = \left| \frac{\Delta}{k} \frac{(k+1)\boldsymbol{\mu}_1 - P}{(tk+t+\frac{\Delta}{k})(k+1)} \right|
$$
$$
= \left| \Delta k \frac{(hk+1)(\boldsymbol{\mu_1} - \boldsymbol{\mu_0})}{(tk+t+\Delta k)(k+1)} \right| \quad (18)
$$
$$
= \left| \Delta k \frac{(hk+1)(\boldsymbol{\mu_1} - \boldsymbol{\mu_0})(tk+1-\Delta k)}{(tk+t+\Delta k)(tk+1-\Delta k)(k+1)} \right|.
$$

We have $|u|$:

$$
|u| = \left| \Delta k \frac{(k-hk)(\boldsymbol{\mu}_1 - \boldsymbol{\mu}_0)(tk+1+\Delta k)}{(tk+t+\Delta k)(tk+1-\Delta k)(k+1)} \right| \quad (19)
$$

Neglecting the common terms of $|v|$ and $|u|$, we only need to compare the following two terms:

$$
\begin{aligned}
|(1+hk)(tk+1-\Delta k)| \\
|(k-hk)(tk+1+\Delta k)|
\end{aligned} \quad (20)
$$

Both of of them are positive because $1 > h > 0.5$ and $(2h - 1)t > \Delta$ . The former term subtract the latter term:

$$1 + (2h + t - \Delta - 1)k + (2th - t - \Delta)k^2 \qquad (21)$$

As $h > 0.5$ and $(2h - 1)t > \Delta$ due to the assumption, eq. (21) is positive. To sum up, $D_{KL-INS} > D_{KL-DEL}$. This conclusion also holds for class 1. We can conclude that the distribution shift caused by insertion is larger.

$\square$

**Synthetic Graph**    We build a graph containing two types of nodes. The features of class $0 \in \mathcal{R}^{10}$ and sampled from $\mathcal{N}(\mathbf{0}, \Lambda_{0.2})$, and the features of class 1 are sampled from $\mathcal{N}(\mathbf{1}, \Lambda_{0.2})$, where $\Lambda_{0.2}$ is a diagonal matrix, and each element is 0.2. There are 150 nodes in class 0 and class 1, respectively, and all nodes of the same class are connected to each other. Thus, this graph is dense and the possibilities to insert and delete is balanced. We conduct MetaAttack to attack this graph, and the results are listed in Table 10. We find that MetaAttack still tends to inset but not delete edges.

Table 10: Share (in %) of edge deletions and insertions by MetaAttack on the synthetic graph.

| Ptb Rate | Insertion | Deletion |
|---|---|---|
| 5% | 814 | 311 |
| 10% | 2087 | 163 |
| 15% | 2961 | 414 |
| 20% | 3704 | 796 |

### A.6 LEVERAGE OF HIGH-DEGREE NODES

Suppose the node features follow distribution like the normal distribution, in which probability density is higher around the mean. In that case, the aggregation can make the high-degree nodes move to the high-density region and reduce the variance. High-density data can be easily classified and is insensitive to noise Zhu et al. (2022a). According to Li et al. (2022), we can trust them more and assign them higher weights during the aggregation:

$$\boldsymbol{h}_i^t = \text{ReLU}\Big( \big( \sum_{j \in \mathcal{N}_i} \frac{(d_i d_j)^{0.5}}{Z} \boldsymbol{h}_j^{t-1} \big) \boldsymbol{W}_\theta^t \Big), \qquad (22)$$

where $d$ is the degree, and $Z$ is a normalization coefficient. This only modifies the aggregation weights in the GCN and can be merged into any Robust GNNs with vanilla GCN. Table 11 demonstrates that assigning high-density data higher weights can indeed improve the robustness of GNNs. This trick just slightly leverages the properties of high-degree nodes. However, one also might apply a more sophisticated method which we leave for future work.

### A.7 HEURISTIC ATTACK

The algorithm of our heuristic attack is shown in Algorithm 1. We first construct the candidate attacking set $\mathcal{C}$ that contains the nodes whose degrees are lower than the average degree. Then we divide $\mathcal{C}$ into $\mathcal{C}_{train}$ and $\mathcal{C}_{test}$. We generate cross-label edges on the graph according to the data split. Specifically, we compute $\lambda_1$, $\lambda_2$, and $\lambda_3$ using Eq. (3). Here we suppose the training set is much smaller than the testing set, *i.e.*, $\lambda_1 < \lambda_3$. The total perturbation is $N_{ptb} = \Delta |\mathcal{E}|$. We inject $N_{ptb} \frac{\lambda_1}{\lambda_1 + \lambda_2}$ cross-label edges into the Train-Train area and $N_{ptb} \frac{\lambda_2}{\lambda_1 + \lambda_2}$ into the Train-Test area. For the nodes in the training set we label them by the pseudo labels predicted by a two layer vanilla GCN.

In Fig. 6 we compare the proposed algorithm with other attack methods on Citeseer and ogbn-arxiv. On Citeseer, our method can achieve comparable performance to MetaAttack, and even better under low perturbation rates.

The average runtime of 10 runs is shown in Table 12. Gradient-based methods need to optimize all possible entries in the **dense** adjacency matrix $\mathbf{A}$, which comes with expensive computation and quadratic space complexity. Our methods and DICE are both rule-based, much faster, and

Table 11: Classification accuracy(%) on dataset Cora and Citeseer attacked by MetaAttack under different perturbation rates (stronger is bold). The asterisk indicates that the GCN part of this model is replaced with Eq. (22). Jaccard Wu et al. (2019) and SimpGCN Jin et al. (2021) are two robust GNNs with vanilla GCN. The data split follows 10%/ 10%/ 80%(train/ val/test).

| Datasets | Ptb rate | GCN(*) | Jaccard(*) | SimpGCN(*) |
|---|---|---|---|---|
| Cora | 0% | **83.56**(82.76) | **81.79**(81.11) | **83.77**(83.64) |
| | 5% | 76.36(**78.17**) | 80.23(**80.57**) | 78.98(**80.45**) |
| | 10% | 71.62(**74.23**) | 74.65(**76.99**) | 75.07(**78.04**) |
| | 15% | 66.37(**70.89**) | 74.29(**76.32**) | 71.42(**75.31**) |
| | 20% | 60.31(**69.59**) | 73.11(**73.42**) | 68.90(**73.29**) |
| Citeseer | 0% | **74.63**(74.00) | **73.64**(73.11) | 74.66(**75.23**) |
| | 5% | 71.13(**72.71**) | 71.15(**71.48**) | 73.54(**74.25**) |
| | 10% | 67.49(**69.17**) | 69.85(**70.53**) | 72.03(**72.90**) |
| | 15% | 61.59(**64.35**) | 67.50(**68.82**) | 69.82(**72.15**) |
| | 20% | 56.26(**60.86**) | 67.01(**67.14**) | 69.59(**71.26**) |

---

**Algorithm 1:** Heuristic Attack

---

**Input:** Graph $\mathcal{G} = \{\mathcal{V}, \mathcal{E}\}$, Labels $\boldsymbol{y}_L$, Perturbation rate $\Delta$
**Output:** The poisoned graph $\mathcal{G}'$

1 $N_{ptb} = \Delta |\mathcal{E}|$;
2 Compute the degrees $\mathbf{d}$ of all nodes and the mean $d_{mean}$;
3 Construct the low-degree node set $\mathcal{C} = \{v_i$ for $v_i$ in $\mathcal{V}$ if $\mathbf{d_i} < d_{mean}\}$ ;
4 Construct $\mathcal{C}_{train}$ that contains training nodes in $\mathcal{C}$ and $\mathcal{C}_{test}$ that contains testing nodes in $\mathcal{C}$ ;
5 Compute $\lambda_1$, $\lambda_2$, and $\lambda_3$ using Eq. (3);
6 $\lambda_{min} = \min(\lambda_1, \lambda_3)$ $\quad\quad\quad\quad\quad\quad$ ▷ Suppose $\lambda_1 < \lambda_3$;
7 $r_1 = \frac{\lambda_1}{\lambda_1 + \lambda_2}$;
8 $r_2 = \frac{\lambda_2}{\lambda_1 + \lambda_2}$;
9 **for** *i=1, ..., $r_1 \cdot N_{ptb}$* **do**
10 $\quad$ Connect two nodes from different class in $\mathcal{C}_{train}$ that are not connected yet.
11 **end**
12 **for** *i=1, ..., $r_2 \cdot N_{ptb}$* **do**
13 $\quad$ Connect one node in $\mathcal{C}_{train}$ and one node in $\mathcal{C}_{test}$ that are from different class and not connected yet.
14 **end**
15 return the poisoned graph $\mathcal{G}'$;

---

space-saving. Our method runs faster than DICE in ogbn-arxiv because ours randomly sample nodes from the candidate set, which is much smaller than the entire node-set.

Our heuristic attack method is coarse-grained and straightforward, and we do not aim to propose a SOTA mode. We try to clarify that by following Tips 1 to 5 and inheriting the tendencies which enlarge the distribution shift, performance close to that of the gradient-based method can be achieved efficiently. More than that, it can be scaled to large graphs.

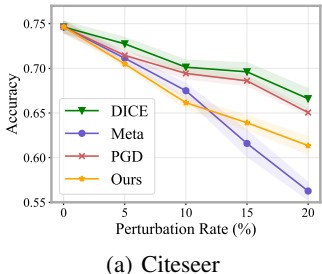
(a) Citeseer

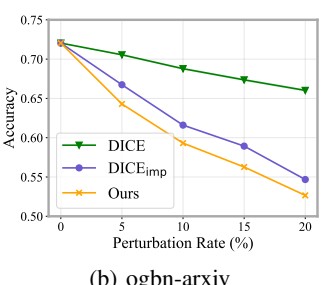
(b) ogbn-arxiv

Figure 6: The adversarial accuracy of GCN attacked by different methods on Citeseer and ogbn-arxiv. PGD and MetaAttack face the OOM problem on ogbn-arxiv.

Table 12: Average time cost (s) of different attack methods on Citeseer and ogbn-arxiv after 10 runs.

| $\Delta$ (Citeseer) | 5% | 10% | 15% | 20% |
|---|---|---|---|---|
| Ours | 0.063 | 0.121 | 0.185 | 0.248 |
| PGD | 7.143 | 7.172 | 7.1650 | 7.186 |
| Meta | 129.227 | 271.354 | 405.466 | 548.913 |
| DICE | 0.043 | 0.044 | 0.046 | 0.047 |
| $\Delta$ (ogbn-arxiv) | 5% | 10% | 15% | 20% |
| Ours | 6.642 | 11.696 | 16.564 | 22.496 |
| PGD | OOM | / | / | / |
| Meta | OOM | / | / | / |
| DICE | 49.243 | 53.521 | 56.213 | 56.755 |

## A.8 IMPROVE ROBUSTNESS VIA SELF-TRAINING

10%/10%/80% is the most commonly used data split in graph adversarial attack, under which effective poisoning attack methods like MetaAttack and PGD tend to attack the training nodes, so the local structure of testing nodes is nearly clean (according to Fig. 5). To summarize, the clean information includes the labels of training nodes and the local structure of test nodes. Thus, a straightforward method to enhance the robustness is self-training Li et al. (2018). We can assign pseudo-labels to the testing nodes and then train the GNN via the pseudo-labels and the clean local structure.

Here we provide a very simple implementation. We use an MLP instead of a GNN to acquire the pseudo-labels because the local structure of training nodes is contaminated. Specifically, we first train an MLP with given labels, then select the predictions with the highest confidence for each class by comparing the softmax scores and add them to a new label set $\mathcal{V}_{psu}$. We finally train a new GCN by computing cross-entropy loss on $\mathcal{V}_{psu}$. This Self-Training Robust GCN (STRG) is described in Algorithm 2. Considering this method from the perspective of data distribution, it actually reduces the distribution shift effectively by training with $p_{test}$. Although the pseudo-labels will introduce some new problems, like label noise, it can be viewed as a trade-off between performance and robustness.

We compare STRG with some baselines and SOTA robust GNNs on two datasets under MetaAttack in Table 13. We set $m = 80$ in these two datasets GNNGuard Zhang & Zitnik (2020) and ProGNN Jin et al. (2020) are two robust GNNs with structure learning. SimpGCN Jin et al. (2021) utilizes a $kNN$ Graph to keep the nodes with similar features close in the representation space and a self-learning regularization to keep the nodes with dissimilar features remote. Elastic Liu et al. (2021a) introduces $\ell_1$-norm to graph signal estimator and proposes elastic message passing which is derived from one-step optimization of such estimator. The local smoothness adaptivity enables the Elastic GNNs robust to structural attacks. STABLE Li et al. (2022) optimizes the graph structures by unsupervised representations learned by contrastive learning. The data split is 10%/10%/80%, and we set the perturbation rate from 0% to 20%. The implementation of these methods follows Appendix A.1, and the hyper-parameter $t$ in STRG is 80.

---

**Algorithm 2:** Self-Training Robust GCN (STRG)

---

**Input:** Graph $\mathcal{G} = \{\mathcal{V}, \mathcal{E}\}$, Labels $\boldsymbol{y}_L$
**Output:** The predictions of testing nodes

---

1  $\mathbf{Z} = \text{MLP}(\mathbf{X}) \in \mathbb{R}^{N \times |\mathcal{C}|}$, $\mathbf{Z}$ is the output of a well-trained MLP;
2  Initial an empty node set $\mathcal{V}_{psu}$;
3  **for** *each class $c_k$* **do**
4  $\quad$ Find the top $m$ nodes in $\mathbf{Z}_{:,k}$;
5  $\quad$ Add them to the $\mathcal{V}_{psu}$;
6  **end**
7  Return the predictions of $f^*$;

---

Table 13: Classification accuracy(%) under different perturbation rates. The top two performance is highlighted in bold and underline.

| Dataset | Ptb Rate | GCN | GNNGuard | ProGNN | SimPGCN | Elastic | STABLE | STRG |
|---------|----------|-----|----------|--------|---------|---------|--------|------|
| Cora | 0% | 83.56±0.25 | 78.52±0.46 | 84.55±0.30 | 83.77±0.57 | 84.76±0.53 | **85.58±0.56** | 82.59±0.65 |
| | 5% | 76.36±0.84 | 77.96±0.54 | 79.84±0.49 | 78.98±1.10 | 82.00±0.39 | 81.40±0.54 | **82.52±0.59** |
| | 10% | 71.62±1.22 | 74.86±0.54 | 74.22±0.31 | 75.07±2.09 | 76.18±0.46 | 80.49±0.61 | **82.34±0.25** |
| | 15% | 66.37±1.97 | 74.15±1.64 | 72.75±0.74 | 71.42±3.29 | 74.41±0.97 | 78.55±0.44 | **79.96±0.39** |
| | 20% | 60.31±1.98 | 72.03±1.11 | 64.40±0.59 | 68.90±3.22 | 69.64±0.62 | **77.80±1.10** | 77.72±0.27 |
| Citeseer | 0% | 74.63±0.66 | 70.07±1.31 | 74.73±0.31 | 74.66±0.79 | 74.86±0.53 | 75.82±0.41 | **76.46±0.67** |
| | 5% | 71.13±0.55 | 69.43±1.46 | 72.88±0.32 | 73.54±0.92 | 73.28±0.59 | 74.08±0.58 | **75.98±0.58** |
| | 10% | 67.49±0.84 | 67.89±1.09 | 69.94±0.45 | 72.03±1.30 | 73.41±0.36 | 73.45±0.40 | **76.56±0.52** |
| | 15% | 61.59±1.46 | 69.14±0.84 | 62.61±0.64 | 69.82±1.67 | 67.51±0.45 | 73.15±0.53 | **76.58±0.84** |
| | 20% | 56.26±0.99 | 69.20±0.78 | 55.49±1.50 | 69.59±3.49 | 65.65±1.95 | 72.76±0.53 | **76.14±0.40** |

We can observe that STRG outperforms other methods under different perturbation rates. In particular, STRG achieves almost complete robustness on the Citeseer. The performance shows no degradation as the perturbation rate rises. Additionally, MLP is much faster than graph models, so this self-training strategy can be easily scaled to large graphs. Here we use GCN as the downstream task classifier, but in fact any GNNs can be merged with it.

We design this method with a hypothesis, *i.e.*, that adversarial edges are mostly located around training nodes, but attackers can perturb $p_{test}$ to make an unbiased model predict the biased data. However, here is also a trade-off between performance and unnoticeabilty for attackers. If the attacker spreads their attacks over the whole graph instead of focusing on the training set, the performance will drop considerably (we will elaborate on this in Appendix A.9). In a nutshell, STRG can successfully defend against effective attack methods, and when an attacker tries to bypass the defense strategy of STRG, the attack will fail.

## A.9 DISCUSSION ON THE DATA SPLIT

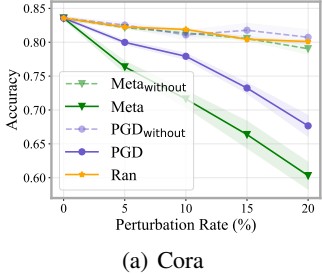
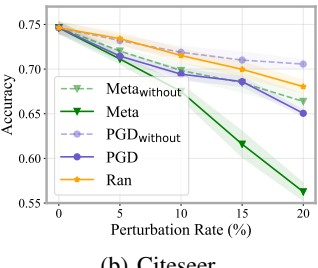

(a) Cora

(b) Citeseer

.

Figure 7: The adversarial accuracy of GCN on Cora and Citeseer. PGD$_{without}$ and Meta$_{without}$ are two variants without the information of data split

Nearly all gradient-based attack methods know the data split by default. However, the importance of it is ignored. Fig. 7 shows that MetaAttack and PGD attack Cora and Citeseer without the information

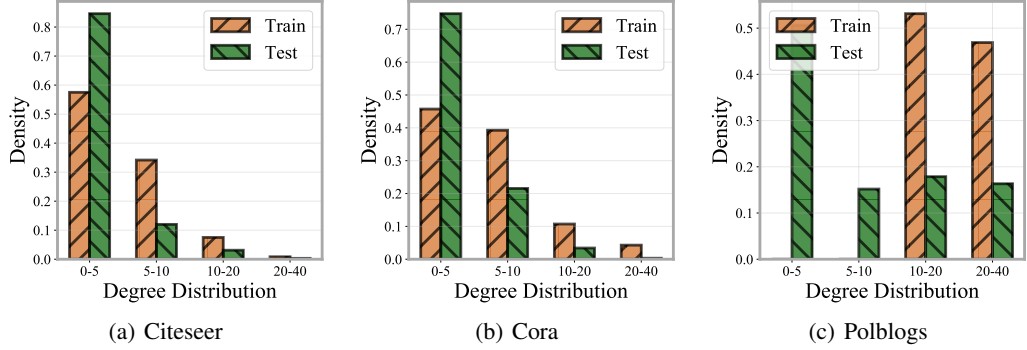

Figure 8: The degree distribution of Citeseer, Cora, and Polblogs attacked by MetaAttack under 5% perturbation. The degrees of training nodes are higher than the testing nodes, which may make the attack noticeable.

Table 14: Performance of GCN attacked by different methods under 5% perturbation where the Public denotes the widely used data split that uses 20 nodes per class as training set.

| Datasets | Cora | | Citeseer | |
|---|---|---|---|---|
| Split | Public | 10%10%80% | Public | 10%10%80% |
| Clean | 81.23±0.31 | 83.56±0.25 | 69.73±0.52 | 74.63±0.66 |
| MetaAttack | 63.79±0.68 | 76.36±0.84 | 52.04±0.43 | 71.13±0.55 |
| $\text{PGD}_{self}$ | 70.16±0.34 | 79.98±0.31 | 58.50±0.26 | 71.45±0.47 |
| FGSM | 71.15±0.56 | 80.37±0.48 | 62.53±0.45 | 72.55±0.61 |

of data split. Data split is necessary when implementing PGD and MetaAttack, so we conduct the attack in a random split, which is inconsistent with the split during testing the classifier. PGD$_{\text{without}}$ and Meta$_{\text{without}}$ are not as effective as the vanilla models, and their performance is even close to the random attack. It is hard to effectively manipulate the prediction without the information of the data split. Meanwhile, attackers and defenders can easily enhance their model by attacking or defending the more vulnerable part of the graph. It is essential to realize that the leakage of the data split can pose a severe security risk.

Moreover, the public split, *i.e.*, 20 nodes per class as the training set, will make the training set significantly small so that $p_{train}$ is easy to be perturbed. We list the performance of poisoning attack methods under different data splits in Table 14. The attack algorithms can easily work on the public split. In addition, such a small training set might make the attack noticeable. As shown in Fig. 8, under a small perturbation rate (*e.g.* 5%), if attackers only modify the local structure around the training nodes, the attack is easily detected.

