# OpenReview forum: "Revisiting Graph Adversarial Attack and Defense From a Data Distribution Perspective"
_ICLR.cc/2023/Conference — ICLR 2023 poster_

### Official Review · Reviewer_Vakj · 2022-10-24

**Confidence:** 3
**Correctness:** 3
**Technical Novelty And Significance:** 3
**Empirical Novelty And Significance:** 3
**Recommendation:** 6

**Clarity, Quality, Novelty And Reproducibility:**

This paper is clear enough and one can follow the train of thought. Novelty is enough. Their experimental results may NOT be verified. There is no code provided.

**Strength And Weaknesses:**

Strength:
- It is an important and interesting problem to investigate.
- Addressing this problem from a distribution perspective is interesting
- Provided tips for attacks and defenses that can be useful for other research works are very interesting.
---------------------------------------------------
Weakness:
- This paper claims to have found out that the adversarial attack is successful because it enlarges the shift between test and training. This is, however, a well-understood result from image adversarial attack and it is obvious.
- This paper emphasizes that the gradient-based attack causes this shift. Could authors please justify this a bit further? Can these results only be applied to gradient-based attacks or is it applicable to different types of attacks?
- it seems like their analysis of distribution shift is limited to the choice of data and the attack.


**Summary Of The Paper:**

This paper explains the reason behind the effectiveness of Gradient-based attacks for poisoning and evasion on semi-supervised node classification tasks on the graph.

**Summary Of The Review:**

This paper has some merits, some results are taken from image adversarial attacks but overall the idea is novel enough. The experimental results are not verifiable. It is recommended that the author submit their code so that reviewers can make sure their results are accurate.

---

> ### Author Response · Authors · 2022-11-06
> **First reply to reviewer Vakj**
>
> We would like to thank you for your insightful comments and helpful suggestions. Let me answer your concerns below. Please let us know if there are any other questions.
>
> >*The experimental results are not verifiable. It is recommended that the authors submit their code so that reviewers can make sure their results are accurate.*
>
> We submit the code of the two proposed algorithms in the rebuttal revision (Supplementary Material). Other experiments are proofs of concept, and they are implemented in Jupyter. If you have questions about the results of them, please point them out, and I will upload the notebook file.
>
> >*This paper claims to have found out that the adversarial attack is successful because it enlarges the shift between test and training. This is, however, a well-understood result from image adversarial attack and it is obvious.*
>
> It is well-known that attacks succeed because they increase the shift between the training set and the testing set in the image adversarial attack, but it is still not clear in graph attack. As is shown in Fig. 2, the training nodes and the testing nodes are in the same graph, so attackers have access to both the training set and the testing set. Training data and testing data in the same graph make previous studies ignore this nature of the attack, increasing the distribution shift. Besides, it is also unknown how the modifications of the structure made by the attack algorithms affect the distribution shift.
>
>  One of our contributions is that we point out the differences between image attack and graph attack. On one hand, graph adversaries can simultaneously attack the training nodes and the testing nodes, and we reveal that the uneven distribution of perturbations effectively enlarges the distribution shift. This is unique in graph attack. On the other hand, formulating the distribution should take the graph structure into account, which is another challenge in graph adversarial attack.
>
> >*This paper emphasizes that the gradient-based attack causes this shift. Could authors please justify this a bit further? Can these results only be applied to gradient-based attacks or is it applicable to different types of attacks?*
>
> The logical chain in this work is that perturbations generated by gradient-based methods are unevenly distributed in the graph, and then we prove that such uneven distribution will cause a larger distribution shift (Theorem 4.1). Most previous attack methods [1, 5, 6, 7, 8] are gradient-based. They rely on computing the gradient of the dense adjacency matrix, so as is shown in our work, they exhibit similar tendencies, and we prove such tendencies effectively increase the distribution shift. For the latter question, as we said, nearly all previous methods are gradient-based, except DICE, a heuristic method aiming at decreasing the homophily ratio of the graph. Thus, we analyze why gradient-based methods significantly outperform DICE from the data distribution perspective in Section 5.2. It is because DICE fails to enlarge the distribution shift.
>
> >*it seems like their analysis of distribution shift is limited to the choice of data and the attack.*
>
> Cora, Citeseer, and Polblogs are three widely used datasets in many graph adversarial attack works [1, 2, 3, 4, 5], and the ogb-arxiv is one of the most representative large graphs. Nearly all the robustness works in graph use the datasets that we use. Frankly speaking, we do not choose any dataset to make our experiments more significant. And for the attacks, we almost cover all the open-source gradient-based attack methods, including MetaAttack, PGD, FGSM, DICE, RANDOM, and PR-BCD.
>
> [1] Daniel Zügner and Stephan Günnemann. Adversarial attacks on graph neural networks via meta learning. In International Conference on Learning Representations (ICLR), 2019.
>
> [2] Wei Jin, Yao Ma, Xiaorui Liu, Xianfeng Tang, Suhang Wang, and Jiliang Tang. Graph structure learning for robust graph neural networks. In KDD, 2020
>
> [3] Simon Geisler, Daniel Zügner, and Stephan Günnemann. Reliable graph neural networks via robust aggregation. In NeurIPS, 2020
>
> [4] Li, Kuan, et al. Reliable Representations Make A Stronger Defender: Unsupervised Structure Refinement for Robust GNN. In KDD, 2022
>
> [5] Daniel Zügner, Amir Akbarnejad, and Stephan Günnemann. Adversarial attacks on neural networks for graph data. In KDD 2018.
>
> [6] Huijun Wu, Chen Wang, Yuriy Tyshetskiy, Andrew Docherty, Kai Lu, and Liming Zhu. Adversarial examples on graph data: Deep insights into attack and defense. In IJCAI, 2019.
>
> [7] Kaidi Xu, Hongge Chen, Sijia Liu, Pin-Yu Chen, Tsui-Wei Weng, Mingyi Hong, and Xue Lin. Topology attack and defense for graph neural networks: An optimization perspective. arXiv preprint arXiv:1906.04214, 2019a.
>
> [8] Simon Geisler, Tobias Schmidt, Hakan  ̧Sirin, Daniel Zügner, Aleksandar Bojchevski, and Stephan Günnemann. Robustness of graph neural networks at scale. In NeurIPS, 2021.

---

> > ### Comment · Reviewer_Vakj · 2022-11-27
> > **Thank you for the response**
> >
> > I'd like to thank the authors for their comprehensive response. I have read other reviews and the responses. I have increased my score to 6: marginally above the threshold.
> >
> > Thank you!

---

> > > ### Author Response · Authors · 2022-11-27
> > > **Thanks!**
> > >
> > > Thank you for your recognition of our work! We would like to discuss this paper if you have any remaining concerns before the end of the discussion period.

---

> ### Author Response · Authors · 2022-11-25
> **Looking forward to your reply**
>
> Dear Reviewer Vakj, we would greatly appreciate it if you could take a look at our reply to your review and let us know if you have any remaining questions. We look forward to addressing any remaining concerns before the end of the discussion period.

---

### Official Review · Reviewer_Lqax · 2022-10-25

**Confidence:** 4
**Correctness:** 4
**Technical Novelty And Significance:** 3
**Empirical Novelty And Significance:** 2
**Recommendation:** 5

**Clarity, Quality, Novelty And Reproducibility:**

The core idea of this paper is clear and easy to understand, the authors also give theoretical and empirical proofs. Besides, this paper provides the experimental detail in the appendix for reproducibility.

**Strength And Weaknesses:**

S1: This paper explains the structural adversarial attack from a distribution perspective, which is seldom discussed in graph adversarial attack.
S2: This paper both theoretically and empirically proves this ‘distribution shift’ phenomenon and revisiting classical attack methods like PGD and MetaAttack.
S2: This paper gives explanations to the existing graph attack observations from a data distribution view.

W1: The distribution analysis from PGD and MetaAttack is similar to that in [1]. Besides, the usage of pseudo labels in the semi-supervised node classification task is also similar. Maybe the authors could give more explanations to the difference of these two works and how this paper improves from [1].
W2: The baseline set of both attack and defense algorithm comparison in Table 10 and Figure 6 is relatively small, which may degrades the effectiveness of proposed methods.

[1] Zhan H, Pei X. Dealing with the unevenness: deeper insights in graph-based attack and defense[J]. Machine Learning, 2022: 1-33.


**Summary Of The Paper:**

This paper discovers a phenomenon that the gradient-based graph structure adversarial attack aims for increase the train-test distribution shift. The authors provide some theoretical proofs and then give explanations to some graph attack observations based on the prove. What’s more, this paper gives some practical tips on robust GNNs and design their attack and defense algorithm based on some tips.

**Summary Of The Review:**

This paper revisits gradient-based graph structure adversarial attack methods from a distribution view. The authors give some theoretical proofs and explanations to graph attack observations. However, the empirical analysis for pgd and meta-attack from a distribution view is somehow being discussed in the previous work and the authors have not mention relevant literature or compare with these works. As a result, I think this paper is marginally above the acceptance threshold.

---

> ### Author Response · Authors · 2022-11-06
> **First reply to reviewer Lqax**
>
> We would like to thank you for your detailed and insightful comments and questions. We have attached a new version of the paper that we hope will clarify the points that you have raised. Let me answer your concerns below. Please let us know if there are any other questions.
>
> >*The distribution analysis from PGD and MetaAttack is similar to that in [1].*
>
> [1] seems a parallel work to ours. It was published on 07 October 2022 after the submission of ICLR, so we missed it. Although [1] is also based on the discovery of uneven distribution of perturbations, they see this phenomenon as a flaw in existing attack methods, and they mainly focus on designing a novel attack algorithm. Different from them, we prefer to study the mechanism behind it, including the reasons for its generation and the impact on the effectiveness of the attack methods. Hence, we formulate the distribution shift in graph adversarial attack and leverage it to analyze other tendencies of gradient-based attack methods, which can help us understand the robustness and vulnerability of GNNs. We also explore which factors affect the location of adversarial edges, e.g., surrogate loss and how the gradient is obtained (Section 4.3). Meanwhile, several tips are proposed, covering all the structure attack aspects, and most of them are not mentioned in [1]. In a nutshell, despite the existence of [1], we believe our work is helpful broadly to the community. The discussion with [1] is added in the updated version (blue font).
>
> >*The baseline set of both attack and defense algorithm comparison in Table 10 and Figure 6 is relatively small, which may degrade the effectiveness of the proposed methods.*
>
> We appreciate your concern and agree that ensuring performance across more baselines is important, but the baselines in Table 10 and Figure 6 are sufficient to express our thoughts.
>
> In Table 10, GCN, Jaccard, and SimpGCN are three methods that contain the vanilla GCN. We show the performance improvement after replacing their renormalization trick with Eq. 22. On one hand, there are not so many methods using the vanilla GCN as the classifier. On the other hand, their performance improvement is already evident enough to show that high-degree nodes can be used to improve robustness.
>
> For Figure 6, MetaAttack is still one of the most powerful attack methods in small graphs. Many works view it as the SOTA and evaluate the robustness of proposed methods under it [2, 3, 4]. Our heuristic method achieves comparable performance to MetaAttack in much less computing time. As we mentioned in the introduction, this algorithm itself is not the focus of this work. We want to provide a simple example of leveraging the tip. What we want to represent here is that, with the advantage of the tips, an easy-implemented heuristic method can perform like the SOTA with much less computing.
>
> [1] Zhan H, Pei X. Dealing with the unevenness: deeper insights in graph-based attack and defense[J]. Machine Learning, 2022: 1-33.
>
> [2] Simon Geisler, Daniel Zügner, and Stephan Günnemann. Reliable graph neural networks via robust aggregation. In NeurIPS, 2020.
>
> [3] Wei Jin, Tyler Derr, Yiqi Wang, Yao Ma, Zitao Liu, and Jiliang Tang. Node similarity preserving graph convolutional networks. In Proceedings of the 14th ACM International Conference on Web Search and Data Mining, pp. 148–156, 2021.
>
> [4] Xiaorui Liu, Wei Jin, Yao Ma, Yaxin Li, Hua Liu, Yiqi Wang, Ming Yan, and Jiliang Tang. Elastic graph neural networks. In International Conference on Machine Learning, pp. 6837–6849. PMLR, 2021.

---

> ### Author Response · Authors · 2022-12-11
> **Reminder: end of the discussion stage**
>
> Dear Reviewer Lqax, since the discussion period will end on 12 Dec, we would greatly appreciate it if you could take a look at our reply to your review and let us know if you have any remaining questions. We look forward to addressing any remaining concerns before the end of the discussion period.
>
> We noticed that you summarized that "this paper is marginally above the acceptance threshold" but gave a score 5 (marginally below the threshold). We're not sure if it was a wrong click or something.

---

### Official Review · Reviewer_Gcf5 · 2022-10-25

**Confidence:** 4
**Correctness:** 3
**Technical Novelty And Significance:** 4
**Empirical Novelty And Significance:** 3
**Recommendation:** 8

**Clarity, Quality, Novelty And Reproducibility:**

Clarity:
The paper is well-written and relatively clear with interesting organizations. Note that there is a typo in Figure 2(b), where the first Testing set should be Training set.

Quality:
The quality of the theoretical analysis is high but the assumptions need further clarification.

Novelty:
Some of the observation is also captured in previous works, but the theoretical foundation has novelty.

Reproducibility:
As the authors claimed, the proposed algorithms are easy to implement, though no codes provided weak the reproducibility.


**Strength And Weaknesses:**

Strengths:
* The paper contains multiple original ideas that are also well-executed and theoretically well-grounded.
* The attack analysis on the behaviors of gradient-based attack methods is novel and could provide insights for further graph robustness research.
* The empirical observations are interesting and the practical tips seem useful for future graph learning attack and defense designs.

Weaknesses:
* The assumption that $p_{train}(\tilde{x}, y)$ and $p_{test}(\tilde{x}, y)$ are the same on clean graphs might be too strong as there are many works on the out-of-distribution analysis on the graphs recently[1]. Perhaps the authors should narrow down the analysis to graphs with specific types to meet this assumption.
* While the empirical results on the impact of $L_{train}$ and $L_{self}$ are interesting, how the pseudo labels affect the theoretical results seems unknown. Meanwhile, why there are no results for MetaAttack for the impact of $L_{train}$ and $L_{self}$?
* Some of the related works are missing. For example, a similar observation on the unevenness distribution is made in [2] and the authors are suggested to discuss the difference with [2] in detail. Meanwhile, some other literature[3,4,5] that analyzes the behaviors of adversarial examples from the spectral perspective are also suggested to be included in the related works.

[1] Out-Of-Distribution Generalization on Graphs: A Survey, ArXiv 2022

[2] Dealing with the unevenness: deeper insights in graph-based attack and defense, Machine Learning 2022

[3] Adversarial Attacks on Node Embeddings via Graph Poisoning, ICML 2019

[4] Not All Low-Pass Filters are Robust in Graph Convolutional Networks, NeurIPS 2021

[5] Adversarial Attack Framework on Graph Embedding Models with Limited Knowledge, TKDE 2022



**Summary Of The Paper:**

This paper thoroughly studies two gradient-based attack methods on graph-structured data, namely MetaAttack and PGD, and reveals their behavior patterns from a data distribution shift perspective. Based on the theoretical analysis regarding training and test data distribution, the authors propose several practical tips from both attack and defense perspectives.
Experiments that are specifically designed for these tips align with the theoretical foundations and show insights for future designs.


**Summary Of The Review:**

In summary, the paper has merits from the theoretical analysis and shows insights for future attack and defense designs on graph learning with practical tips. Meanwhile, my concorns are mainly from two aspects:
* The assumption for train and test data distributions need further clarification. The discussion on the impact of pseudo labels versus real labels can be improved.
* Some of related works should be included.

---

> ### Author Response · Authors · 2022-11-06
> **First reply to Gcf5**
>
> Thank you for your review, as well as your positive feedback on the insights and theoretical foundation. We have attached a new version of the paper that we hope will clarify the points that you have raised. Let me answer your questions below. Please let us know if there are any other questions.
>
> >*The assumption that $p_{train}$ and $p_{test} $are the same on clean graphs. Perhaps the authors should narrow down the analysis to graphs with specific types to meet this assumption.*
>
> We agree that OOD is another critical problem; we adopt the IID assumption for two main reasons. First, the datasets that we use are citation graphs and blog graphs. OOD works like [1] convert them into OOD datasets in some syntactic ways, so they do not view them as original OOD datasets. Second, we make the IID assumption for the clean data here because if we take OOD into account, the study of robustness will become extremely complex. Anyway, it is still valuable to consider OOD, we are willing to do this in future works. Thanks for this suggestion of clarifying the specific types of graphs that meet this assumption. We will specify the graph types that we use in the Section A.1.
>
> >*While the empirical results on the impact of $L_{train}$ and $L_{self}$ are interesting, how the pseudo labels affect the theoretical results seems unknown. Meanwhile, why there are no results for MetaAttack for the impact of $L_{train}$ and $L_{self}$?*
>
> This is an insightful question, let us clarify how pseudo-labels work. Attacks with pseudo labels behave similarly to those with ground-truth labels because  (see the table below; Meta-true means Meta-self uses the ground-truth labels to generate perturbations instead of pseudo labels). We guess it is because, in homophilous graphs, the pseudo labels are generally accurate. Thus, attacks with $L_{self}$ will focus on modifying the local structure of the testing nodes to make the predictions away from the ground-truth labels. In many cases, the size of the testing set is relatively large, so it is hard to increase the distribution shift by modifying the testing structure according to Theorem 4.1. This way, methods like $PGD_{self}$ fail.
>
> |Ptb rate|Meta-self|Meta-true|
> |-|-|-|
> |5%|76.36|75.84|
> |10%|71.62|71.33|
> |15%|66.37|64.42|
> |20%|60.31|58.81|
>
>
> We provide the results for MetaAttack for the impact of $L_{train}$ and $L_{self}$ in Table 7 in A.4. Metaself is the only one that always attacks the more easily perturbed distribution in $p_{train}$ and $p_{test}$.
>
> >*Some of the related works are missing.*
>
> [2] seems a parallel work to ours. It was published on 07 October 2022, which was posted after we submitted this, so we missed it. We add the comparison to it in related works. Although [2] is also based on the discovery of uneven distribution of perturbations, they see this phenomenon as a flaw in existing attack methods, and they mainly focus on designing a novel black-box attack algorithm. Different from them, we prefer to study the mechanism behind it, including the reasons for its generation and the impact on the effectiveness of the attack methods. Hence, we formulate the distribution shift (Eq. 6) in graph adversarial attack and leverage it to analyze other tendencies of gradient-based attack methods (Section 5), which can provide some theoretical guidance and help us understand the robustness and vulnerability of GNNs. We also explore which factors affect the location of adversarial edges, e.g., surrogate loss, and how the gradient is obtained (Section 4.3). We find $L_{self}$ and $L_{train}$ can control the distribution of the perturbations, and the meta-gradient enables the attack algorithm to adaptively adjust the attack distribution according to the size of the training set (A.4). Besides, [2] only study two gradient-based poisoning attack methods, i.e., PGD and MetaAttack, but we explore 4, including an evasion attack. Meanwhile, several tips are proposed, covering all the structure attack aspects, and most of them are not mentioned in [2]. The similarity between [2] and ours is that we both find the phenomenon of uneven distribution of perturbations, but the other parts are very different. In a nutshell, despite the existence of [2], we believe our work is helpful broadly to the community.
>
> We add [3, 4, 5] in the related work in Rebuttal Revision.
>
> [1] GOOD: A Graph Out-of-Distribution Benchmark, NeurIPS, 2022.
>
> [2] Dealing with the unevenness: deeper insights in graph-based attack and defense, Machine Learning 2022.
>
> [3] Adversarial Attacks on Node Embeddings via Graph Poisoning, ICML 2019.
>
> [4] Not All Low-Pass Filters are Robust in Graph Convolutional Networks, NeurIPS 2021.
>
> [5] Adversarial Attack Framework on Graph Embedding Models with Limited Knowledge, TKDE 2022.

---

> > ### Comment · Reviewer_Gcf5 · 2022-11-17
> > **Thank you**
> >
> > Many thanks for your further feedback, especially on the results for MetaAttack for the impact of $L_{train}$ and $L_{self}$ and added discussion of related works, which are helpful.
> >
> > I am happy to increase my score a little bit. Hopefully, the authors could include these points as well as the discussion in the updated version.

---

> > > ### Author Response · Authors · 2022-11-17
> > > **Thanks for increasing your rating.**
> > >
> > > Many thanks for kindly increasing the score. We really appreciate it. All the discussions about the related work as well as the appended results are included in our updated version (blue font). We would like to discuss this paper further if you need.

---

> ### Author Response · Authors · 2022-11-14
> **Looking forward to your reply**
>
> Dear Reviewer Gcf5, as the discussion period is half over, we would greatly appreciate it if you could take a look at our reply to your review and let us know if you have any remaining questions. We look forward to addressing any remaining concerns before the end of the discussion period.

---

### Official Review · Reviewer_3B87 · 2022-10-28

**Confidence:** 3
**Correctness:** 3
**Technical Novelty And Significance:** 3
**Empirical Novelty And Significance:** 2
**Recommendation:** 5

**Clarity, Quality, Novelty And Reproducibility:**

The writing of this paper can be improved.

This paper provides different aspects to understand how does attackers work on graph data.

**Strength And Weaknesses:**


Strength
1. The research problem is very important and this paper provides different aspects to understand how does attackers work on graph data.
2. They summarize some useful tricks to conduct adversarial attacks and defense.




There are some concerns regarding this paper.
1. Distribution Shift is unclear to me. It seems they mainly focus on edges among train-train, train-test, test-test. It would be better if you could motivate this well and provide better explanations.
It's still hard to understand the distribution shift for attacking.



2. This paper focuses on adversarial attacks on structure attacks, which limit their applications in various tasks and can not be well generalized to many tasks.



3. As for the tips on both attack and defense, some of them have been found in Adversarial examples on graph data: Deep insights into attack and defense-2019.
It would be better if this paper could provide more insights into graph data from both structure and feature aspects.


**Summary Of The Paper:**

This paper tries to have a better understanding of adversarial attacks on graphs. They revisit graph adversarial attack from a data distribution perspective and formulate the distribution shift: the adversarial edges are not uniformly distributed on the graph.
Then, they provide an explanation for the effectiveness of the gradient-based attack method from a data distribution perspective and revisit both poisoning attack and evasion attack. Next, they provide several practical tips on both attack and defense and meanwhile leverage them to improve existing attack and defense methods.

**Summary Of The Review:**


Strength
1. The research problem is very important and this paper provides different aspects to understand how does attackers work on graph data.
2. They summarize some useful tricks to conduct adversarial attacks and defense.




There are some concerns regarding this paper.
1. Distribution Shift is unclear to me. It seems they mainly focus on edges among train-train, train-test, test-test. It would be better if you could motivate this well and provide better explanations.
It's still hard to understand the distribution shift for attacking.



2. This paper focuses on adversarial attacks on structure attacks, which limit their applications in various tasks and can not be well generalized to many tasks.



3. As for the tips on both attack and defense, some of them have been found in Adversarial examples on graph data: Deep insights into attack and defense-2019.
It would be better if this paper could provide more insights into graph data from both structure and feature aspects.

---

> ### Author Response · Authors · 2022-11-06
> **First reply to reviewer 3B87**
>
> Thank you for the helpful and insightful feedback, and please see below for our response to your concerns. Please let us know if there are any other questions.
>
> >*Distribution Shift is unclear to me. It seems they mainly focus on edges among train-train, train-test, test-test. It would be better if you could motivate this well and provide better explanations. It's still hard to understand the distribution shift for attacking.*
>
> We find the perturbations are unevenly distributed in the graph, namely the adversarial edges are generated around training nodes. This phenomenon inspires us to study this problem from the perspective of distribution shift from two aspects. First, the concentration of adversarial edges around the training nodes can lead to large differences in the local structure of training nodes and testing nodes, causing a significant feature distribution shift after aggregation. Additionally, the attack algorithm treats the local structure of training nodes and testing nodes differently, and the distribution shift considers the differences between the training set and the testing set. The two are naturally related.
>
> >*This paper focuses on adversarial attacks on structure attacks*
>
> We agree that studying the perturbations in features is an important problem, and we are willing to work on this in the future. In this work, we focus on structural perturbations like many previous representative works [1, 2, 3, 4] in this field.  Most works focus on structural attack because this is unique in graph tasks. Besides, we cover all the aspects of structure attack, including poisoning attack, evasion attack, and defense, so the scope is indeed large.
>
> >*The differences between our tips and the methods in [5].*
>
> In [5], the authors proposed a defense algorithm and an attack algorithm. The defense model Jaccard is based on the homophily assumption, and they improve the robustness of GNN by removing the edges which link two dissimilar nodes. Our tips for defense models mostly focus on the distribution of perturbations on the graph, so they are nothing like Jaccard in [5]. And for the attack algorithm, they choose where to attack by computing integrated gradients. Our tips for attack are based on increasing the distribution shift between the training set and the testing set.
>
> [5] was an early and influential work, concentrating more on solving the problem of how to attack and defend on graphs, namely designing specific attack and defense algorithms, but we prefer to explain the effectiveness of the attack methods. With the advantage of our claims and observations, we can enhance existing models and design new ones. Our attack method and STRG are two proofs of concept, which simply follow the corresponding tips and then can achieve comparable performance. To summarize, all our tips are based on the distribution shift, and none of them are mentioned in [5].
>
>
> [1] Daniel Zügner and Stephan Günnemann. Adversarial attacks on graph neural networks via meta learning. ICLR, 2019.
>
> [2] Simon Geisler, Tobias Schmidt, Hakan ¸Sirin, Daniel Zügner, Aleksandar Bojchevski, and Stephan Günnemann. Robustness of graph neural networks at scale. In NeurIPS, 2021.
>
> [3] Yulin Zhu, Yuni Lai, Kaifa Zhao, Xiapu Luo, Mingquan Yuan, Jian Ren, and Kai Zhou. Binarizedattack: Structural poisoning attacks to graph-based anomaly detection. In ICDE, 2022b.
>
> [4] Chang, Heng, Yu Rong, Tingyang Xu, Yatao Bian, Shiji Zhou, Xin Wang, Junzhou Huang, and Wenwu Zhu. Not all low-pass filters are robust in graph convolutional networks. In NeurIPS, 2021.
>
> [5] Huijun Wu, Chen Wang, Yuriy Tyshetskiy, Andrew Docherty, Kai Lu, and Liming Zhu. Adversarial examples on graph data: Deep insights into attack and defense. In IJCAI, 2019.

---

> ### Author Response · Authors · 2022-12-11
> **Reminder: end of the discussion stage**
>
> Dear Reviewer 3B87, since the discussion period will end on 12 Dec, we would greatly appreciate it if you could take a look at our reply to your review and let us know if you have any remaining questions. We look forward to addressing any remaining concerns before the end of the discussion period.

---

### Public Comment · ~Yongqiang_Chen1 · 2022-11-07
**Missing discussion with a highly related work and some questions**

Hi, this is really an interesting work that discovers the distribution shifts in terms of structural information in a semi-supervised node classification task. The tips are informative with plentiful theoretical and empirical support.

We found that the authors seem to miss the discussion with a highly related work [1], where we formally study the distribution shifts during the graph injection attack in terms of node feature information, i.e., homophily distribution before and after attacks.

Maybe I missed something, but it seems that the authors are studying the evasion attacks in the transductive semi-supervised node classification. It seems this setup would violate the definition of the transductive setting where the model is able to access all of the nodes and edges during training, as the adversary would modify the nodes and edges during the testing. Although it seems not to invalidate the main conclusions and contributions of the paper, could the authors offer some explanations for studying this setup?



**References**

[1] Chen et al., Understanding and Improving Graph Injection Attack by Promoting Unnoticeability, ICLR 2022.

---

> ### Author Response · Authors · 2022-11-07
> **Reply to Yongqiang Chen**
>
> Hi Yongqiang, thanks for your interest in our paper. I read [1] on the openreview about this time last year. It is an interesting work, and I was impressed by the discussion of the changes in homophily (Fig. 2c). Another very important theoretical foundation in [1] is that Graph Injection Attack (GIA) can cause more damage than Graph Modification Attack (GMA) with equal or fewer budgets. The proposed method HAO can simultaneously degrade the performance of GNNs and maintain homophily unnoticeability.
>
> Let me answer your questions below.
>
> >*We found that the authors seem to miss the discussion with a highly related work [1], where we formally study the distribution shifts during the graph injection attack in terms of node feature information, i.e., homophily distribution before and after attacks.*
>
> In this paper, one major contribution is that we explore where and why the perturbations are located in the original graph, i.e., the train-train, train-test, and test-test. This is unique in GMA, so we do not discuss the GIA papers [1, 2, 3]. The homophily shift discussed in [1] is whole-graph-level, focusing on the homophily change in the entire graph. We focus more on part-graph-level, namely which part of the graph is significantly changed (see Fig.8). We discover that the unnoticeability is more damaged in the local graph around the training nodes. Anyway, we think it is necessary to discuss unnoticeability in both the full and partial graphs. We will add the discussion with [1] in A.9, where we talk about the unnoticeability.
>
> >*Maybe I missed something, but it seems that the authors are studying the evasion attacks in the transductive semi-supervised node classification. It seems this setup would violate the definition of the transductive setting where the model is able to access all of the nodes and edges during training, as the adversary would modify the nodes and edges during the testing. Although it seems not to invalidate the main conclusions and contributions of the paper, could the authors offer some explanations for studying this setup?*
>
> This work covers three aspects of graph adversarial attack, poisoning attack, evasion attack, and defense. The evasion attack setting may not be completely consistent with the transductive setting, as the graph is modified after training. However, it is still valuable to study the robustness under such a setting. For example, in online trading graphs, the graph structure keeps changing over time, but the model might be trained in one previous timestamp. Some users want to conceal their true intentions by deliberately linking other nodes. Your discussion on the transductive setting is reasonable, and here we call it transductive evasion attack for mainly two reasons. First, the evasion attacks in the transductive semi-supervised node classification are broadly explored in GMA [4, 5, 6, 7]. In addition, as we said, such a setting is still valuable, and we can hardly say that it is identical to inductive or transductive. The graph is changed after attacking, while the structure information and features of testing data are indeed used to train the model. Hence, we consider it more like a transductive task due to the limited modifications to the graph.
>
> [1] Chen et al., Understanding and Improving Graph Injection Attack by Promoting Unnoticeability, ICLR 2022.
>
> [2] Zou et al., TDGIA: Effective Injection Attacks on Graph Neural Networks, KDD 2021
>
> [3] Ju, Mingxuan, et al. "Black-box Node Injection Attack for Graph Neural Networks." arXiv preprint arXiv:2202.09389 (2022).
>
> [4] Daniel Z¨ugner, Amir Akbarnejad, and Stephan G¨unnemann. Adversarial attacks on neural networks for graph data. In KDD, 2018.
>
> [5] Simon Geisler, Daniel Zügner, and Stephan Günnemann. Reliable graph neural networks via robust aggregation. In NeurIPS, 2020.
>
> [6] Liang Chen, Jintang Li, Qibiao Peng, Yang Liu, Zibin Zheng, and Carl Yang. Understanding structural vulnerability in graph convolutional networks. arXiv preprint arXiv:2108.06280, 2021.
>
> [7] Simon Geisler, Tobias Schmidt, Hakan ¸ Sirin, Daniel Zügner, Aleksandar Bojchevski, and Stephan Günnemann. Robustness of graph neural networks at scale. In NeurIPS, 2021.

---

> > ### Public Comment · ~Yongqiang_Chen1 · 2022-11-08
> > **Thank you for the detailed explanation**
> >
> > Hi authors, thank you very much for your detailed explanation. They help me understand more about the results and contributions of this work : )

---

### Author Response · Authors · 2022-12-12
**Summary of the discussion period**

Many thanks to all the reviewers for their careful reading and insightful reviews. We have updated the paper accordingly and uploaded the rebuttal revision. As the discussion phase is ending in a few hours, we summarize our responses and the feedback of reviewers below.

**Concerns summary:**

Reviewer Gcf5 and Vakj have replied to our responses and confirmed that their concerns are well addressed. The main concern of reviewer Lqax is the same as Gcf5 about missing a related paper [1]. As we said in the rebuttal, [1] was published and posted on 07 October 2022 after the submission of ICLR, so we missed it. We add the comparison with it in the related work, and reviewer Gcf5 approved our statement that the similarity between [2] and ours is that we both find the phenomenon of uneven distribution of perturbations, but the other parts are very different. Despite the existence of [1], our work is still helpful broadly to the community.

Regarding the concerns of reviewer 3B87, we believe our response can well address them. Much of the literature in graph adversarial attack only focuses on one specific type of structural attack or defense, e.g., the poisoning attack or evasion attack [2, 3, 4, 5, 6]. We systematically discuss poisoning attack, evasion attack, and defense, so the scope is indeed large (Weakness 2). [3] was an early and influential work, concentrating more on solving the problem of how to attack and defend on graphs, the perspective of distribution shift and our tips are not mentioned in it (Weakness 3).

**Another thing worth noting** is that reviewer Lqax summarized that "this paper is marginally above the acceptance threshold" in the initial review but gave a score 5 (marginally below the threshold). We're not sure if it was a wrong click or something.

**Contributions:** The main contribution of this work is conceptual but not algorithmic. The two proposed attack and defense algorithms are just simple implementations of our tips. We aim to show that we can achieve comparable performance to SOTA with much lower computation cost by following the tips. Also, experimental results can demonstrate that the tips can be used to improve existing attack and defense models. We believe our discoveries, formulations, and tips can be helpful for future research in this domain.

[1] Zhan H, Pei X. Dealing with the unevenness: deeper insights in graph-based attack and defense[J]. Machine Learning, 2022: 1-33.

[2] Daniel Zügner and Stephan Günnemann. Adversarial attacks on graph neural networks via meta learning. ICLR, 2019.

[3] Huijun Wu, Chen Wang, Yuriy Tyshetskiy, Andrew Docherty, Kai Lu, and Liming Zhu. Adversarial examples on graph data: Deep insights into attack and defense. In IJCAI, 2019.

[4] Simon Geisler, Daniel Zügner, and Stephan Günnemann. Reliable graph neural networks via robust aggregation. In NeurIPS, 2020

[5] Wei Jin, Tyler Derr, Yiqi Wang, Yao Ma, Zitao Liu, and Jiliang Tang. Node similarity preserving graph convolutional networks. In Proceedings of the 14th ACM International Conference on Web Search and Data Mining, pp. 148–156, 2021.

[6] Simon Geisler, Tobias Schmidt, Hakan ¸ Sirin, Daniel Zügner, Aleksandar Bojchevski, and Stephan Günnemann. Robustness of graph neural networks at scale. In NeurIPS, 2021.

---

### Decision · Program_Chairs · 2023-01-20

**Decision:**

Accept: poster

**Justification For Why Not Higher Score:**

Some of the insights have been known before, thus, the novelty is not very high.

**Justification For Why Not Lower Score:**

This is a borderline paper and, if required, the score can be lowered to reject.

**Metareview: Summary, Strengths And Weaknesses:**

The authors investigate adversarial attacks to graph neural networks. Specifically, they aim to understand how adversarial (structure) perturbations affect the underlying data distribution. The paper provides some interesting insights about the behavior of adversarial attacks along with "tips" how to perform good attacks and defense (though, these tips and insights are not entirely new and I strongly recommend the authors to add corresponding citations to literature in case these "tips" have already been discovered). Overall, I recommend acceptance as a poster.

**Note From Pc:**

if the above contains the word "oral" or "spotlight" please see: "oral" presentation means -> notable-top-5% and "spotlight" means -> notable-top-25%. As stated in our emails, we are disassociating presentation type from AC recommendations